# Vector-output ReLU Neural Network Problems are Copositive Programs: Convex Analysis of Two Layer Networks and Polynomial-time Algorithms

**Arda Sahiner, Tolga Ergen, John Pauly & Mert Pilanci**
Department of Electrical Engineering
Stanford University
{sahiner, ergen, pauly, pilanci}@stanford.edu

## ABSTRACT

We describe the convex semi-infinite dual of the two-layer vector-output ReLU neural network training problem. This semi-infinite dual admits a finite dimensional representation, but its support is over a convex set which is difficult to characterize. In particular, we demonstrate that the non-convex neural network training problem is equivalent to a finite-dimensional convex copositive program. Our work is the first to identify this strong connection between the global optima of neural networks and those of copositive programs. We thus demonstrate how neural networks implicitly attempt to solve copositive programs via semi-nonnegative matrix factorization, and draw key insights from this formulation. We describe the first algorithms for provably finding the global minimum of the vector output neural network training problem, which are polynomial in the number of samples for a fixed data rank, yet exponential in the dimension. However, in the case of convolutional architectures, the computational complexity is exponential in only the filter size and polynomial in all other parameters. We describe the circumstances in which we can find the global optimum of this neural network training problem exactly with soft-thresholded SVD, and provide a copositive relaxation which is guaranteed to be exact for certain classes of problems, and which corresponds with the solution of Stochastic Gradient Descent in practice.

## 1 INTRODUCTION

In this paper, we analyze vector-output two-layer ReLU neural networks from an optimization perspective. These networks, while simple, are the building blocks of deep networks which have been found to perform tremendously well for a variety of tasks. We find that vector-output networks regularized with standard weight-decay have a convex semi-infinite strong dual–a convex program with infinitely many constraints. However, this strong dual has a finite parameterization, though expressing this parameterization is non-trivial. In particular, we find that expressing a vector-output neural network as a convex program requires taking the convex hull of completely positive matrices. Thus, we find an intimate, novel connection between neural network training and copositive programs, i.e. programs over the set of completely positive matrices (Anjos & Lasserre, 2011). We describe algorithms which can be used to find the global minimum of the neural network training problem in polynomial time for data matrices of fixed rank, which holds for convolutional architectures. We also demonstrate under certain conditions that we can provably find the optimal solution to the neural network training problem using soft-thresholded Singular Value Decomposition (SVD). In the general case, we introduce a relaxation to parameterize the neural network training problem, which in practice we find to be tight in many circumstances.

### 1.1 RELATED WORK

Our analysis focuses on the optima of finite-width neural networks. This approach contrasts with certain approaches which have attempted to analyze infinite-width neural networks, such as the

Neural Tangent Kernel (Jacot et al., 2018). Despite advancements in this direction, infinite-width neural networks do not exactly correspond to their finite-width counterparts, and thus this method of analysis is insufficient for fully explaining their success (Arora et al., 2019).

Other works may attempt to optimize neural networks with assumptions on the data distribution. Of particular interest is (Ge et al., 2018), which demonstrates that a polynomial number of samples generated from a planted neural network model is sufficient for extracting its parameters using tensor methods, assuming the inputs are drawn from a symmetric distribution. If the input distribution to a simple convolutional neural network with one filter is Gaussian, it has also been shown that gradient descent can find the global optimum in polynomial time (Brutzkus & Globerson, 2017). In contrast to these works, we seek to find general principles for learning two-layer ReLU networks, regardless of the data distribution and without planted model assumptions.

Another line of work aims to understand the success of neural networks via implicit regularization, which analyzes how models trained with Stochastic Gradient Descent (SGD) find solutions which generalize well, even without explicit control of the optimization objective (Gunasekar et al., 2017; Neyshabur et al., 2014). In contrast, we consider the setting of explicit regularization, which is often used in practice in the form of weight-decay, which regularizes the sum of squared norms of the network weights with a single regularization parameter $\beta$, which can be critical for neural network performance (Golatkar et al., 2019).

Our approach of analyzing finite-width neural networks with a fixed training dataset has been explored for networks with a scalar output (Pilanci & Ergen, 2020; Ergen & Pilanci, 2020a;d). In fact, our work here can be considered a generalization of these results. We consider a ReLU-activation two-layer network $f : \mathbb{R}^d \to \mathbb{R}^c$ with $m$ neurons:

$$f(\boldsymbol{x}) = \sum_{j=1}^{m} (\boldsymbol{x}^\top \boldsymbol{u}_j)_+ \boldsymbol{v}_j^\top \qquad (1)$$

where the function $(\cdot)_+ = \max(0, \cdot)$ denotes the ReLU activation, $\{\boldsymbol{u}_j \in \mathbb{R}^d\}_{j=1}^m$ are the first-layer weights of the network, and $\{\boldsymbol{v}_j \in \mathbb{R}^c\}_{j=1}^m$ are the second-layer weights. In the scalar-output case, the weights $\boldsymbol{v}_j$ are scalars, i.e. $c = 1$. Pilanci & Ergen (2020) find that the neural network training problem in this setting corresponds to a finite-dimensional convex program.

However, the setting of scalar-output networks is limited. In particular, this setting cannot account for tasks such as multi-class classification or multi-dimensional regression, which are some of the most common uses of neural networks. In contrast, the vector-output setting is quite general, and even greedily training and stacking such shallow vector-output networks can match or even exceed the performance of deeper networks on large datasets for classification tasks (Belilovsky et al., 2019). We find that this important task of extending the scalar case to the vector-output case is an exceedingly non-trivial task, which generates novel insights. Thus, generalizing the results of Pilanci & Ergen (2020) is an important task for a more complete knowledge of the behavior of neural networks in practice.

Certain works have also considered technical problems which arise in our analysis, though in application they are entirely different. Among these is analysis into cone-constrained PCA, as explored by Deshpande et al. (2014) and Asteris et al. (2014). They consider the following optimization problem

$$\max_{\boldsymbol{u}} \boldsymbol{u}^\top \boldsymbol{R} \boldsymbol{u}$$
$$\text{s.t } \boldsymbol{X} \boldsymbol{u} \geq 0;\ \|\boldsymbol{u}\|_2 = 1 \qquad (2)$$

This problem is in general considered NP-hard. Asteris et al. (2014) provide an exponential algorithm which runs in $\mathcal{O}(n^d)$ time to find the exact solution to (2), where $\boldsymbol{X} \in \mathbb{R}^{n \times d}$ and $\boldsymbol{R} \in \mathcal{S}^d$ is a symmetric matrix. We leverage this result to show that the optimal value of the vector-output neural network training problem can be found in the worst case in exponential time with respect to $r := \textbf{rank}(\mathbf{X})$, while in the case of a fixed-rank data matrix our algorithm is polynomial-time. In particular, convolutional networks with fixed filter sizes (e.g., $3 \times 3 \times m$ convolutional kernels) correspond to the fixed-rank data case (e.g., $r = 9$). In search of a polynomial-time approximation

to (2), Deshpande et al. (2014) evaluate a relaxation of the above problem, given as

$$\max_{\boldsymbol{U}} \langle \boldsymbol{R}, \boldsymbol{U} \rangle$$
$$\text{s.t } \boldsymbol{X}\boldsymbol{U}\boldsymbol{X}^\top \geq 0; \ \mathbf{tr}(\boldsymbol{U}) = 1; \ \boldsymbol{U} \succeq 0 \tag{3}$$

While the relaxation not tight in all cases, the authors find that in practice it works quite well for approximating the solution to the original optimization problem. This relaxation, in particular, corresponds to what we call a *copositive relaxation*, because it consists of a relaxation of the set $\mathcal{C}_{PCA} = \{\boldsymbol{u}\boldsymbol{u}^\top : \|\boldsymbol{u}\|_2 = 1, \boldsymbol{X}\boldsymbol{u} \geq 0\}$. When $\boldsymbol{X} = \boldsymbol{I}$ and the norm constraint is removed, $\mathcal{C}_{PCA}$ is the set of completely positive matrices (Dür, 2010). Optimizing over the set of completely positive matrices is NP-hard, as is optimizing over its convex hull:

$$\mathcal{C} := \mathbf{conv}\{\boldsymbol{u}\boldsymbol{u}^\top : \boldsymbol{X}\boldsymbol{u} \geq 0\}$$

Thus, optimizing over $\mathcal{C}$ is a convex optimization problem which is nevertheless NP-hard. Various relaxations to $\mathcal{C}$ have been proposed, such as the copositive relaxation used by (Deshpande et al., 2014) above:

$$\tilde{\mathcal{C}} := \{\boldsymbol{U} : \boldsymbol{U} \succeq 0; \ \boldsymbol{X}\boldsymbol{U}\boldsymbol{X}^\top \geq 0\}$$

In fact, this relaxation is tight, given that $\boldsymbol{u} \in \mathbb{R}^d$ and $d \leq 4$ (Burer, 2015; Kogan & Berman, 1993). However, $\mathcal{C} \subset \tilde{\mathcal{C}}$ for $d \geq 5$, so the copositive relaxation provides a lower bound in the general case. These theoretical results prove insightful for understanding the neural network training objective.

## 1.2 Contributions

- We find the semi-infinite convex strong dual for the vector-output two-layer ReLU neural network training problem, and prove that it has a finite-dimensional exact convex optimization representation.

- We establish a new connection between vector-output neural networks, copositive programs and cone-constrained PCA problems, yielding new insights into the nature of vector-output neural network training, which extend upon the results of the scalar-output case.

- We provide methods that globally solve the vector-output neural network training problem in polynomial time for data matrices of a fixed rank, but for the full-rank case, the complexity is necessarily exponential in $d$ assuming $\mathcal{P} \neq \mathcal{NP}$.

- We provide conditions on the training data and labels with which we can find a closed-form expression for the optimal weights of a vector-output ReLU neural network using soft-thresholded SVD.

- We propose a copositive relaxation to establish a heuristic for solving the neural network training problem. This copositive relaxation is often tight in practice.

## 2 Preliminaries

In this work, we consider fitting labels $\boldsymbol{Y} \in \mathbb{R}^{n \times c}$ from inputs $\boldsymbol{X} \in \mathbb{R}^{n \times d}$ with a two layer neural network with ReLU activation and $m$ neurons in the hidden layer. This network is trained with weight decay regularization on all of its weights, with associated parameter $\beta > 0$. For some general loss function $\ell(f(\boldsymbol{X}), \boldsymbol{Y})$, this gives us the non-convex primal optimization problem

$$p^* = \min_{\substack{\boldsymbol{u}_j \in \mathbb{R}^d \\ \boldsymbol{v}_j \in \mathbb{R}^c}} \frac{1}{2}\ell(f(\boldsymbol{X}), \boldsymbol{Y}) + \frac{\beta}{2}\sum_{j=1}^{m}\left(\|\boldsymbol{u}_j\|_2^2 + \|\boldsymbol{v}_j\|_2^2\right) \tag{4}$$

In the simplest case, with a fully-connected network trained with squared loss[1], this becomes:

$$p^* = \min_{\substack{\boldsymbol{u}_j \in \mathbb{R}^d \\ \boldsymbol{v}_j \in \mathbb{R}^c}} \frac{1}{2}\|\sum_{j=1}^{m}(\boldsymbol{X}\boldsymbol{u}_j)_+\boldsymbol{v}_j^\top - \boldsymbol{Y}\|_F^2 + \frac{\beta}{2}\sum_{j=1}^{m}\left(\|\boldsymbol{u}_j\|_2^2 + \|\boldsymbol{v}_j\|_2^2\right) \tag{5}$$

---

[1]Appendix A.6 contains extensions to general convex loss functions.

However, alternative models can be considered. In particular, for example, Ergen & Pilanci (2020d) consider two-layer CNNs with global average pooling, for which we can define the patch matrices $\{\boldsymbol{X}_k\}_{k=1}^{K}$, which define the patches which individual convolutions operate upon. Then, the vector-output neural network training problem with global average pooling becomes

$$p_{conv}^* = \min_{\substack{\boldsymbol{u}_j \in \mathbb{R}^d \\ \boldsymbol{v}_j \in \mathbb{R}^c}} \frac{1}{2}\|\sum_{k=1}^{K}\sum_{j=1}^{m}(\boldsymbol{X}_k\boldsymbol{u}_j)_+\boldsymbol{v}_j^\top - \boldsymbol{Y}\|_F^2 + \frac{\beta}{2}\sum_{j=1}^{m}\left(\|\boldsymbol{u}_j\|_2^2 + \|\boldsymbol{v}_j\|_2^2\right) \tag{6}$$

We will show that in this convolutional setting, because the rank of the set of patches $\boldsymbol{M} := [\boldsymbol{X}_1, \boldsymbol{X}_2, \cdots \boldsymbol{X}_K]^\top$ cannot exceed the filter size of the convolutions, there exists an algorithm which is polynomial in all problem dimensions to find the global optimum of this problem. We note that such matrices typically exhibit rapid singular value decay due to spatial correlations, which may also motivate replacing it with an approximation of much smaller rank.

In the following section, we will demonstrate how the vector-output neural network problem has a convex semi-infinite strong dual. To understand how to parameterize this semi-infinite dual in a finite fashion, we must introduce the concept of hyper-plane arrangements. We consider the set of diagonal matrices

$$\mathcal{D} := \{\text{diag}(\mathbf{1}_{\boldsymbol{X}\boldsymbol{u}\geq 0}) : \|\boldsymbol{u}\|_2 \leq 1\}$$

This is a finite set of diagonal matrices, dependent on the data matrix $\boldsymbol{X}$, which indicate the set of possible arrangement activation patterns for the ReLU non-linearity, where a value of 1 indicates that the neuron is active, while 0 indicates that the neuron is inactive. In particular, we can enumerate the set of sign patterns as $\mathcal{D} = \{\boldsymbol{D}_i\}_{i=1}^{P}$, where $P$ depends on $\boldsymbol{X}$ but is in general bounded by

$$P \leq 2r\left(\frac{e(n-1)}{r}\right)^r$$

for $r := \textbf{rank}(\boldsymbol{X})$ (Pilanci & Ergen, 2020; Stanley et al., 2004). Note that for a fixed rank $r$, such as in the convolutional case above, $P$ is polynomial in $n$. Using these sign patterns, we can completely characterize the range space of the first layer after the ReLU:

$$\{(\boldsymbol{X}\boldsymbol{u})_+ : \|\boldsymbol{u}\|_2 \leq 1\} = \{\boldsymbol{D}_i\boldsymbol{X}\boldsymbol{u} : \|\boldsymbol{u}\|_2 \leq 1, \ (2\boldsymbol{D}_i - \boldsymbol{I})\boldsymbol{X}\boldsymbol{u} \geq 0, \ i \in [P]\}$$

We also introduce a class of data matrices $\boldsymbol{X}$ for which the analysis of scalar-output neural networks simplifies greatly, as shown in (Ergen & Pilanci, 2020b). These matrices are called *spike-free* matrices. In particular, a matrix $\boldsymbol{X}$ is called spike-free if it holds that

$$\{(\boldsymbol{X}\boldsymbol{u})_+ : \|\boldsymbol{u}\|_2 \leq 1\} = \{\boldsymbol{X}\boldsymbol{u} : \|\boldsymbol{u}\|_2 \leq 1\} \cap \mathbb{R}_+^n \tag{7}$$

When $\boldsymbol{X}$ is spike-free, then, the set of sign patterns $\mathcal{D}$ reduces to a single sign pattern, $\mathcal{D} = \{I\}$, because of the identity in (7). The set of spike-free matrices includes (but is not limited to) diagonal matrices and whitened matrices for which $n \leq d$, such as the output of Zero-phase Component Analysis (ZCA) whitening. The setting of whitening the data matrix has been shown to improve the performance of neural networks, even in deeper settings where the whitening transformation is applied to batches of data at each layer (Huang et al., 2018). We will see that spike-free matrices provide polynomial-time algorithms for finding the global optimum of the neural network training problem in both $n$ and $d$ (Ergen & Pilanci, 2020b), though the same does not hold for vector-output networks.

## 2.1 WARM-UP: SCALAR-OUTPUT NETWORKS

We first present strong duality results for the scalar-output case, i.e. the case where $c = 1$.

**Theorem** (Pilanci & Ergen, 2020) *There exists an $m^* \leq n + 1$ such that if $m \geq m^*$, for all $\beta > 0$, the neural network training problem (5) has a convex semi-infinite strong dual, given by*

$$p^* = d^* := \max_{\boldsymbol{z}: \ |\boldsymbol{z}^\top(\boldsymbol{X}\boldsymbol{u})_+|\leq\beta \ \forall\|\boldsymbol{u}\|_2\leq 1} -\frac{1}{2}\|\boldsymbol{z} - \boldsymbol{y}\|_2^2 + \frac{1}{2}\|\boldsymbol{y}\|_2^2 \tag{8}$$

*Furthermore, the neural network training problem has a convex, finite-dimensional strong bi-dual, given by*

$$p^* = \min_{\substack{(2\boldsymbol{D}_i - \boldsymbol{I})\boldsymbol{X}\boldsymbol{w}_i \geq 0 \ \forall i \in [P] \\ (2\boldsymbol{D}_i - \boldsymbol{I})\boldsymbol{X}\boldsymbol{v}_i \geq 0 \ \forall i \in [P]}} \frac{1}{2}\|\sum_{i=1}^{P}\boldsymbol{D}_i\boldsymbol{X}(\boldsymbol{w}_i - \boldsymbol{v}_i) - \boldsymbol{y}\|_2^2 + \beta\sum_{i=1}^{P}\|\boldsymbol{w}_i\|_2 + \|\boldsymbol{v}_i\|_2 \tag{9}$$

This is a convex program with $2dP$ variables and $2nP$ linear inequalities. Solving this problem with standard interior point solvers thus has a complexity of $\mathcal{O}(d^3 r^3 (\frac{n}{d})^{3r})$, which is thus exponential in $r$, but for a fixed rank $r$ is polynomial in $n$.

In the case of a spike-free $\boldsymbol{X}$, however, the dual problem simplifies to a single sign pattern constraint $\boldsymbol{D}_1 = \boldsymbol{I}$. Then the convex strong bi-dual becomes (Ergen & Pilanci, 2020b)

$$p^* = \min_{\substack{\boldsymbol{Xw} \geq 0 \\ \boldsymbol{Xv} \geq 0}} \frac{1}{2} \|\boldsymbol{X}(\boldsymbol{w} - \boldsymbol{v}) - \boldsymbol{y}\|_2^2 + \beta \Big( \|\boldsymbol{w}\|_2 + \|\boldsymbol{v}\|_2 \Big) \tag{10}$$

This convex problem has a much simpler form, with only $2n$ linear inequality constraints and $2d$ variables, which therefore has a complexity of $\mathcal{O}(nd^2)$. We will see that the results of scalar-output ReLU neural networks are a specific case of the vector-output case.

## 3 STRONG DUALITY

### 3.1 CONVEX SEMI-INFINITE DUALITY

**Theorem 1** *There exists an $m^* \leq nc + 1$ such that if $m \geq m^*$, for all $\beta > 0$, the neural network training problem (5) has a convex semi-infinite strong dual, given by*

$$p^* = d^* := \max_{\boldsymbol{Z}: \, \|\boldsymbol{Z}^\top (\boldsymbol{Xu})_+\|_2 \leq \beta \, \forall \|\boldsymbol{u}\|_2 \leq 1} -\frac{1}{2} \|\boldsymbol{Z} - \boldsymbol{Y}\|_F^2 + \frac{1}{2} \|\boldsymbol{Y}\|_F^2 \tag{11}$$

*Furthermore, the neural network training problem has a convex, finite-dimensional strong bi-dual, given by*

$$p^* = \min_{\boldsymbol{V}_i \in \mathcal{K}_i \, \forall i \in [P]} \frac{1}{2} \| \sum_{i=1}^{P} \boldsymbol{D}_i \boldsymbol{X} \boldsymbol{V}_i - \boldsymbol{Y} \|_F^2 + \beta \sum_{i=1}^{P} \|\boldsymbol{V}_i\|_* \tag{12}$$

*for convex sets $\mathcal{K}_i$*

$$\mathcal{K}_i := \mathbf{conv}\{\boldsymbol{u}\boldsymbol{g}^\top : \, (2\boldsymbol{D}_i - \boldsymbol{I})\boldsymbol{X}\boldsymbol{u} \geq 0, \, \|\boldsymbol{g}\|_2 \leq 1\} \tag{13}$$

The strong dual given in (11) is convex, albeit with infinitely many constraints. In contrast, (12) is a convex problem has finitely many constraints. This convex model learns a sparse set of locally linear models $\boldsymbol{V}_i$ which are constrained to be in a convex set, for which group sparsity and low-rankness over hyperplane arrangements is induced by the sum of nuclear-norms penalty. The emergence of the nuclear norm penalty is particularly interesting, since similar norms have also been used for rank minimization problems (Candès & Tao, 2010; Recht et al., 2010), proposed as implicit regularizers for matrix factorization models (Gunasekar et al., 2017), and draws similarities to nuclear norm regularization in multitask learning (Argyriou et al., 2008; Abernethy et al., 2009), and trace Lasso (Grave et al., 2011). We note the similarity of this result to that from Pilanci & Ergen (2020), whose formulation is a special case of this result with $c = 1$, where $\mathcal{K}_i$ reduce to

$$\mathcal{K}_i = \{\boldsymbol{u} : (2\boldsymbol{D}_i - I)\boldsymbol{X}\boldsymbol{u} \geq 0\} \cup \{-\boldsymbol{u} : (2\boldsymbol{D}_i - I)\boldsymbol{X}\boldsymbol{u} \geq 0\}$$

from which we can obtain the convex program presented by Pilanci & Ergen (2020). Further, this result extends to CNNs with global average pooling, which is discussed in Appendix A.3.2.

**Remark 1.1** *It is interesting to observe that the convex program (12) can be interpreted as a piecewise low-rank model that is partitioned according to the set of hyperplane arrangements of the data matrix. In other words, a two-layer ReLU network with vector output is precisely a linear learner over the features $[\boldsymbol{D}_1 \boldsymbol{X}, \cdots \boldsymbol{D}_P \boldsymbol{X}]$, where convex constraints and group nuclear norm regularization $\sum_{i=1}^{P} \|\boldsymbol{V}_i\|_*$ is applied to the linear model weights. In the case of the CNNs, the piecewise low-rank model is over the smaller dimensional patch matrices $\{\boldsymbol{X}_k\}_{k=1}^{K}$, which result in significantly fewer hyperplane arrangements, and therefore, fewer local low-rank models.*

## 3.2 Provably Solving the Neural Network Training Problem

In this section, we present a procedure for minimizing the convex program as presented in (12) for general output dimension $c$. This procedure relies on Algorithm 5 for cone-constrained PCA from (Asteris et al., 2014), and the Frank-Wolfe algorithm for constrained convex optimization (Frank et al., 1956). Unlike SGD, which is a heuristic method applied to a non-convex training problem, this approach is built upon results of convex optimization and provably finds the global minimum of the objective. In particular, we can solve the problem in epigraph form,

$$p^* = \min_{\substack{t \geq 0 \\ \sum_{i=1}^P \|V_i\|_* \leq t}} \min_{V_i \in \mathcal{K}_i \ \forall i \in [P]} \frac{1}{2} \| \sum_{i=1}^P D_i X V_i - Y \|_F^2 + \beta t \tag{14}$$

where we can perform bisection over $t$ in an outer loop to determine the overall optimal value of (12). Then, we have the following algorithm to solve the inner minimization problem of (14):
**Algorithm 1:**

1. Initialize $\{V_i^{(0)}\}_{i=1}^P = 0$.
2. For steps $k$:
   (a) For each $i \in [P]$ solve the following subproblem:

   $$s_i^{(k)} = \max_{\substack{\|u\|_2 \leq 1 \\ \|g\|_2 \leq 1 \\ (2D_i - I) X u \geq 0}} \langle D_i X u g^\top, Y - \sum_{j=1}^P D_j X V_j^{(k)} \rangle$$

   And define the pairs $\{(u_i, g_i)\}_{i=1}^P$ to be the argmaxes of the above subproblems. This is is a form of semi-nonnegative matrix factorization (semi-NMF) on the residual at step $k$ (Ding et al., 2008). It can be solved via cone-constrained PCA in $\mathcal{O}(n^r)$ time where $r = \mathbf{rank}(X)$.
   (b) For the semi-NMF factorization obtaining the largest objective value, $i^* := \arg\max_i s_i^{(k)}$, form $M_{i^*}^{(k)} = u_{i^*} g_{i^*}^\top$. For all other $i \neq i^*$, simply let $M_i^{(k)} = 0$.
   (c) For step size $\alpha^{(k)} \in (0,1)$, update

   $$V_i^{(k+1)} = (1 - \alpha^{(k)}) V_i^{(k)} + t \alpha^{(k)} M_i^{(k)}$$

The derivations for the method and complexity of Algorithm 1 are found in Appendix A.4. We have thus described a Frank-Wolfe algorithm which provably minimizes the convex dual problem, where each step requires a semi-NMF operation, which can be performed in $\mathcal{O}(n^r)$ time.

## 3.3 Spike-free Data Matrices and Closed-Form Solutions

As discussed in Section 2, if $X$ is spike-free, the set of sign partitions is reduced to the single partition $D_1 = I$. Then, the convex program (12) becomes

$$\min_{V \in \mathbf{conv}\{ug^\top : Xu \geq 0, \|g\|_2 \leq 1\}} \frac{1}{2} \|XV - Y\|_F^2 + \beta \|V\|_* \tag{15}$$

This problem can also be solved with Algorithm 1. However, the asymptotic complexity of this algorithm is unchanged, due to the cone-constrained PCA step. If the constraint on $V$ were removed, (15) would be identical to optimizing a linear-activation network. However, additional cone constraint on $V$ allows for a more complex representation, which demonstrates that even in the spike-free case, a ReLU-activation network is quite different from a linear-activation network.

Recalling that whitened data matrices where $n \leq d$ are spike-free, for a further simplified class of data and label matrices, we can find a closed-form expression for the optimal weights.

**Theorem 2** *Consider a whitened data matrix $X \in \mathbb{R}^{n \times d}$ where $n \leq d$, and labels $Y$ with SVD of $X^\top Y = \sum_{i=1}^c \sigma_i a_i b_i^\top$. If the left-singular vectors of $X^\top Y$ satisfy $X a_i \geq 0 \ \forall i \in \{i : \sigma_i > \beta\}$, there exists a closed-form solution for the optimal $V^*$ to problem (15), given by*

$$V^* = \sum_{i=1}^c (\sigma_i - \beta)_+ a_i b_i^\top \tag{16}$$

The resulting model is a soft-thresholded SVD of $\boldsymbol{X}^\top \boldsymbol{Y}$, which arises as the solution of maximum-margin matrix factorization (Srebro et al., 2005). The scenario that all the left singular vectors of $\boldsymbol{X}^\top \boldsymbol{Y}$ satisfy the affine constraints $\boldsymbol{X}\boldsymbol{a}_i \geq 0 \ \forall i$ occurs when the all of the left singular vectors $\boldsymbol{Y}$ are nonnegative, which is the case for example when $\boldsymbol{Y}$ is a one-hot-encoded matrix. In this scenario where the left-singular vectors of $\boldsymbol{X}^\top \boldsymbol{Y}$ satisfy $\boldsymbol{X}\boldsymbol{a}_i \geq 0 \ \forall i \in \{i : \sigma_i > \beta\}$, we note that the ReLU constraint on $\boldsymbol{V}^*$ is not active, and therefore, the solution of the ReLU-activation network training problem is identical to that of the linear-activation network. This linear-activation setting has been well-studied, such as in matrix factorization models by (Cabral et al., 2013; Li et al., 2017), and in the context of implicit bias of dropout (Mianjy et al., 2018; Mianjy & Arora, 2019). This theorem thus provides a setting in which ReLU-activation and linear-activation networks perform identically.

## 4 NEURAL NETWORKS AND COPOSITIVE PROGRAMS

### 4.1 AN EQUIVALENT COPOSITIVE PROGRAM

We now present an alternative representation of the neural network training problem with squared loss, which has ties to copositive programming.

**Theorem 3** *For all $\beta > 0$, the neural network training problem (5) has a convex strong dual, given by*

$$p^* = \min_{\boldsymbol{U}_i \in \mathcal{C}_i \ \forall i \in [P]} \frac{1}{2}\mathbf{tr}\left(\boldsymbol{Y}^\top\left(\boldsymbol{I} + 2\sum_{i=1}^P (\boldsymbol{D}_i\boldsymbol{X})\boldsymbol{U}_i(\boldsymbol{D}_i\boldsymbol{X})^\top\right)^{-1}\boldsymbol{Y}\right) + \beta^2 \sum_{i=1}^P \mathbf{tr}(\boldsymbol{U}_i) \quad (17)$$

*for convex sets $\mathcal{C}_i$, given by*

$$\mathcal{C}_i := \mathbf{conv}\{\boldsymbol{u}\boldsymbol{u}^\top : (2\boldsymbol{D}_i - \boldsymbol{I})\boldsymbol{X}\boldsymbol{u} \geq 0\} \quad (18)$$

This is a minimization problem with a convex objective over $P$ sets of convex, completely positive cones–a copositive program, which is NP-hard. There exists a cutting plane algorithm solves this problem in $\mathcal{O}(n^r)$, which is polynomial in $n$ for data matrices of rank $r$ (see Appendix A.5). This formulation provides a framework for viewing ReLU neural networks as implicit copositive programs, and we can find conditions during which certain relaxations can provide optimal solutions.

### 4.2 A COPOSITIVE RELAXATION

We consider the copositive relaxation of the sets $\mathcal{C}_i$ from (17). We denote this set

$$\tilde{\mathcal{C}}_i := \{\boldsymbol{U} : \boldsymbol{U} \succeq 0, \ (2\boldsymbol{D}_i - \boldsymbol{I})\boldsymbol{X}\boldsymbol{U}\boldsymbol{X}^\top(2\boldsymbol{D}_i - \boldsymbol{I}) \geq 0\}$$

In general, $\mathcal{C}_i \subseteq \tilde{\mathcal{C}}_i$, with equality when $d \leq 4$ (Kogan & Berman, 1993; Dickinson, 2013). We define the relaxed program as

$$d_{cp}^* := \min_{\boldsymbol{U}_i \in \tilde{\mathcal{C}}_i \ \forall i \in [P]} \frac{1}{2}\mathbf{tr}\left(\boldsymbol{Y}^\top\left(\boldsymbol{I} + 2\sum_{i=1}^P (\boldsymbol{D}_i\boldsymbol{X})\boldsymbol{U}_i(\boldsymbol{D}_i\boldsymbol{X})^\top\right)^{-1}\boldsymbol{Y}\right) + \beta^2 \sum_{i=1}^P \mathbf{tr}(\boldsymbol{U}_i) \quad (19)$$

Because of the enumeration over sign-patterns, this relaxed program still has a complexity of $\mathcal{O}(n^r)$ to solve, and thus does not improve upon the asymptotic complexity presented in Section 3.

### 4.3 SPIKE-FREE DATA MATRICES

If $\boldsymbol{X}$ is restricted to be spike-free, the convex program (19) becomes

$$d_{cp}^* := \min_{\substack{\boldsymbol{U} \succeq 0 \\ \boldsymbol{X}\boldsymbol{U}\boldsymbol{X}^\top \geq 0}} \frac{1}{2}\mathbf{tr}\left(\boldsymbol{Y}^\top\left(\boldsymbol{I} + 2\boldsymbol{X}\boldsymbol{U}\boldsymbol{X}^\top\right)^{-1}\boldsymbol{Y}\right) + \beta^2 \mathbf{tr}(\boldsymbol{U}) \quad (20)$$

With spike-free data matrices, the copositive relaxation presents a heuristic algorithm for the neural network training problem which is polynomial in both $n$ and $d$. This contrasts with the exact formulations of (12) and (17), for which the neural network training problem is exponential even for a spike-free $\boldsymbol{X}$. Table 1 summarizes the complexities of the neural network training problem.

Table 1: Complexity of global optimization for two-layer ReLU networks with scalar and vector outputs. Best known upper-bounds are shown where $n$ is the number of samples, $d$ is the dimension of the samples and $r$ is the rank of the training data. Note that for convolutional networks, $r$ is the size of a single filter, e.g., a convolutional layer with a kernel size of $3 \times 3$ corresponds to $r = 9$.

|  | Scalar-output | Vector-output (exact) | Vector-output (relaxation) |
|---|---|---|---|
| Spike-free $\boldsymbol{X}$ | $\mathcal{O}(nd^2)$ | $\mathcal{O}(n^r)$ | $\mathcal{O}(n^2 d^4)$ |
| General $\boldsymbol{X}$ | $\mathcal{O}((\frac{n}{d})^{3r})$ | $\mathcal{O}(n^r(\frac{n}{d})^{3r})$ | $\mathcal{O}((\frac{n}{d})^{3r})$ |

## 5 EXPERIMENTS

### 5.1 DOES SGD ALWAYS FIND THE GLOBAL OPTIMUM FOR NEURAL NETWORKS?

While SGD applied to the non-convex neural network training objective is a heuristic which works quite well in many cases, there may exist pathological cases where SGD fails to find the global minimum. Using Algorithm 1, we can now verify whether SGD find the global optimum. In this experiment, we present one such case where SGD has trouble finding the optimal solution in certain circumstances. In particular, we generate random inputs $\boldsymbol{X} \in \mathbb{R}^{25 \times 2}$, where the elements of $\boldsymbol{X}$ are drawn from an i.i.d standard Gaussian distribution: $\boldsymbol{X}_{i,j} \sim \mathcal{N}(0,1)$. We then randomly initialize a data-generator neural network $f$ with 100 hidden neurons and and an output dimension of 5, and generate labels $\boldsymbol{Y} = f(\boldsymbol{X}) \in \mathbb{R}^{25 \times 5}$ using this model. We then attempt to fit these labels using a neural network and squared loss, with $\beta = 10^{-2}$. We compare the results of training this network for 5 trials with 10 and 50 neurons to the global optimum found by Algorithm 1. In this circumstance, with 10 neurons, none of the realizations of SGD converge to the global optimum as found by Algorithm 1, but with 50 neurons, the loss is nearly identical to that found by Algorithm 1.

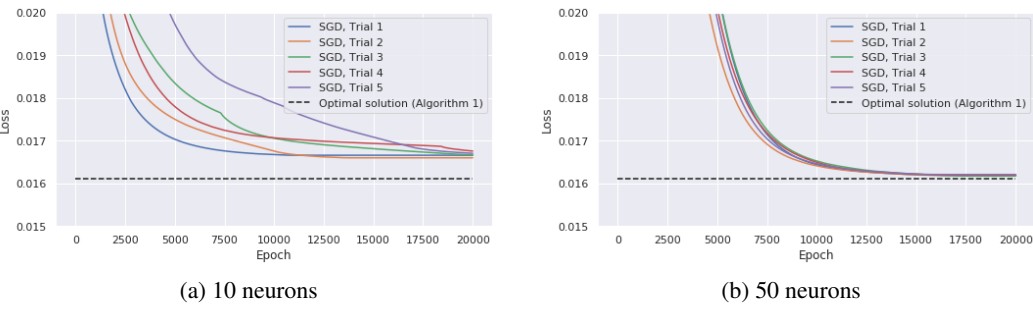

(a) 10 neurons  (b) 50 neurons

Figure 1: As the number of neurons increases, the solution of SGD approaches the optimal value.

### 5.2 MAXIMUM-MARGIN MATRIX FACTORIZATION

In this section, we evaluate the performance of the soft-thresholded SVD closed-form solution presented in Theorem 2. In order to evaluate this method, we take a subset of 3000 points from the CIFAR-10 and CIFAR-100 datasets (Krizhevsky et al., 2009). For each dataset, we first de-mean the data matrix $\boldsymbol{X} \in \mathbb{R}^{3000 \times 3072}$, then whiten the data-matrix using ZCA whitening. We seek to fit one-hot-encoded labels representing the class labels from these datasets. In Fig. 2, we observe that the soft-thresholded SVD method from Theorem 2 finds the same solution as SGD in far shorter time. Appendix A.1.4 contains further details of this experiment.

### 5.3 EFFECTIVENESS OF THE COPOSITIVE PROGRAM

In this section, we compare the objective values obtained by SGD, Algorithm 1, and the copositive program defined in (17). We use an artificially-generated spiral dataset, with $\boldsymbol{X} \in \mathbb{R}^{60 \times 2}$ and 3

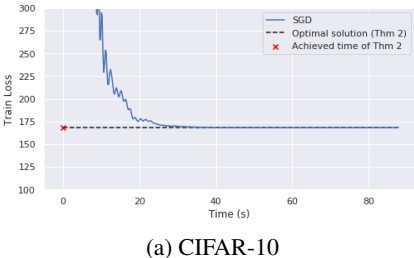 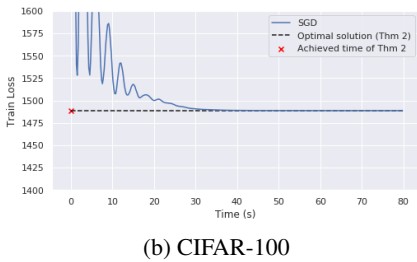

(a) CIFAR-10          (b) CIFAR-100

Figure 2: The maximum-margin SVD from Theorem 2 provides the closed-form solution for the optimal value of the neural network training problem for whitened CIFAR-10 and CIFAR-100.

classes (see Fig. 3(a) for an illustration). In this case, since $d \leq 4$, we note that the copositive relaxation in (19) is tight. Across different values of $\beta$, we compare the solutions found by these three methods. As shown in Fig. 3, the copositive relaxation, the solution found by SGD, and the solution found by Algorithm 1 all coincide with the same loss across various values of $\beta$. This verifies our theoretical proofs of equivalence of (5), (12), and (19).

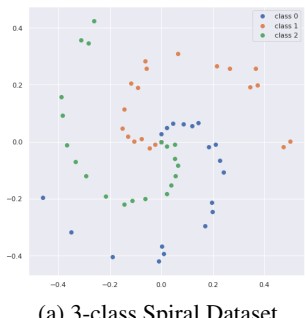 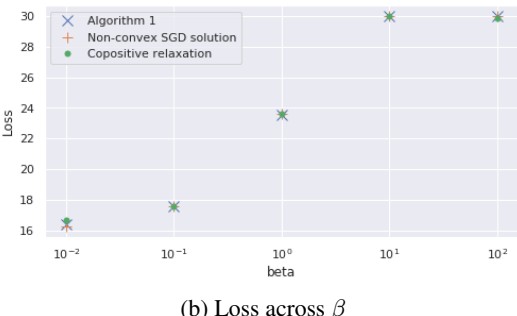

(a) 3-class Spiral Dataset          (b) Loss across $\beta$

Figure 3: Spiral classification: SGD (1000 neurons), Algorithm 1, and copositive relaxation (19).

## 6   CONCLUSION

We studied the vector-output ReLU neural network training problem, and designed the first algorithms for finding the global optimum of this problem, which are polynomial-time in the number of samples for a fixed data rank. We found novel connections between this vector-output ReLU neural network problem and a variety of other problems, including semi-NMF, cone-constrained PCA, soft-thresholded SVD, and copositive programming. Of particular interest is extending these results to deeper networks, which would further explain the performance of neural networks as they are often used in practice. One such method to extend the results in this paper to deeper networks is to greedily train and stack two-layer networks to create one deeper network, which has shown to mimic the performance of deep networks trained end-to-end. Some preliminary results for convex program equivalents of deeper training problems are presented under whitened input data assumptions in (Ergen & Pilanci, 2020c). Another interesting research direction is investigating efficient relaxations of our vector output convex programs for larger scale simulations, which have been studied in (Bartan & Pilanci, 2019; Ergen & Pilanci, 2019b;a; d'Aspremont & Pilanci, 2020). Furthermore, landscapes of vector output neural networks and dynamics of gradient descent type methods can be analyzed by leveraging our results. In (Lacotte & Pilanci, 2020), an analysis of the landscape for scalar output networks based on the convex formulation was given which establishes a direct mapping between the non-convex and convex objective landscapes. Finally, our copositive programming and semi-NMF representations of ReLU networks can be used to develop more interpretable neural models. An investigation of scalar output convex neural models for neural image reconstruction was given in (Sahiner et al., 2020).

ACKNOWLEDGEMENTS

This work was partially supported by the National Science Foundation under grants IIS-1838179 and ECCS-2037304, the National Institutes of Health under grants R01EB009690 and R01EB0026136, Facebook Research, Adobe Research and Stanford SystemX Alliance.

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

# A APPENDIX

## A.1 ADDITIONAL EXPERIMENTAL DETAILS

All neural networks in the experiments were trained using the Pytorch deep learning library (Paszke et al., 2019), using a single NVIDIA GeForce GTX 1080 Ti GPU. Algorithm 1 was trained using a CPU with 256 GB of RAM, as was the maximum-margin matrix factorization. Unless otherwise stated, the Frank-Wolfe method from Algorithm 1 used a step size of $\alpha^{(k)} = \frac{2}{2+k}$, and all methods were trained to minimize squared loss.

Unless otherwise stated, the neural networks were trained until full training loss convergence with SGD with a momentum parameter of 0.95 and a batch size the size of the training set (i.e. full-batch gradient descent), and the learning rate was decremented by a factor of 2 whenever the training loss reached a plateau. The initial learning rate was set as high as possible without causing the training to diverge. All neural networks were initialized with Kaiming uniform initialization (He et al., 2015).

### A.1.1 ADDITIONAL EXPERIMENT: COPOSITIVE RELAXATION WHEN $d \geq 4$

The copositive relaxation for the neural network training problem described in (19) is not guaranteed to exactly correspond to the objective when $d \geq 4$. However, we find that in practice, this relaxation is tight even in such settings. To demonstrate such an instance, we consider the problem of generating images from noise.

In particular, we initialize $\boldsymbol{X}$ element-wise from an i.i.d standard Gaussian distribution. To analyze the spike-free setting, we whitened $\boldsymbol{X}$ using ZCA whitening. Then, we attempted to fit images $\boldsymbol{Y}$ from the MNIST handwritten digits dataset (LeCun et al., 1998) and CIFAR-10 dataset (Krizhevsky et al., 2009) respectively. From each dataset, we select 100 random images with 10 samples from each class and flatten them into vectors, to form $\boldsymbol{Y}_{MNIST} \in \boldsymbol{Y}^{100 \times 784}$ and $\boldsymbol{Y}_{CIFAR} \in \boldsymbol{Y}^{100 \times 3072}$. We allow the noise inputs to have the same shape as the output. Clearly, in these cases, with $d = 784$ and $d = 3072$ respectively, the copositive relaxation (20) is not guaranteed to correspond exactly to the neural network training optimum.

However, we find across a variety of regularization parameters $\beta$, that the solution found by SGD and this copositive relaxation exactly correspond, as demonstrated in Figure 4.

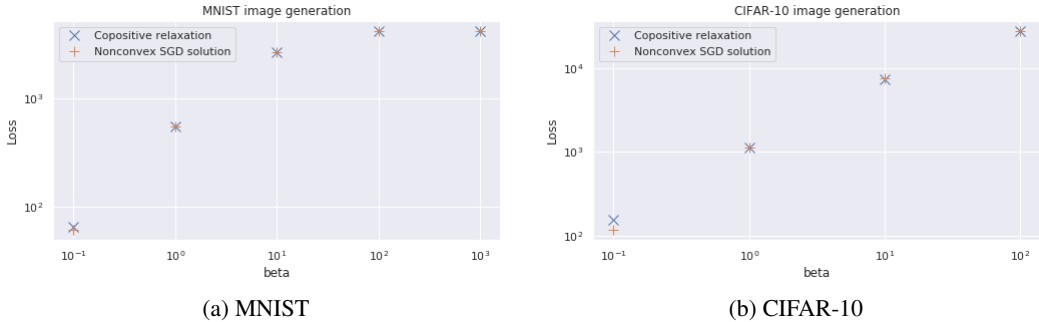

|            |            |
|:----------:|:----------:|
| (a) MNIST  | (b) CIFAR-10 |

Figure 4: The copositive relaxation (20) and solution found by SGD nearly correspond for almost all $\beta$.

While for the lowest value of $\beta$, the copositive relaxation does not exactly correspond with the value obtained by SGD, we note that we showed the objective value of the copositive relaxation to be a *lower bound* of the neural network training objective–meaning that the differences seen in this plot are likely due to a numerical optimization issue, rather than a fundamental one. Non-convex SGD was trained for 60,000 epochs with 1000 neurons with a learning rate of $5 \times 10^{-5}$, while the copositive relaxation was trained using Adam (Kingma & Ba, 2014) with the Geotorch library for constrained optimization and manifold optimization for deep learning in PyTorch, which allowed us

to express the PSD constraint, with an additional hinge loss to penalize the violations of the affine constraints. This copositive relaxation was trained for 60,000 epochs with a learning rate of $10^{-2}$ for CIFAR-10 and $4 \times 10^{-2}$ for MNIST, and $\beta_1 = 0.9$, $\beta_2 = 0.999$ and $\epsilon = 10^{-8}$ as parameters for Adam.

### A.1.2 ADDITIONAL EXPERIMENT: COMPARING ReLU ACTIVATION TO LINEAR ACTIVATION IN THE CASE OF SPIKE-FREE MATRICES

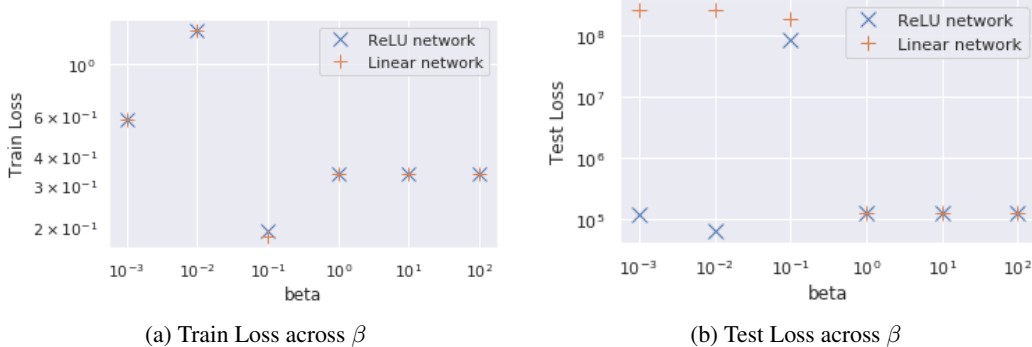

(a) Train Loss across $\beta$       (b) Test Loss across $\beta$

Figure 5: Comparing train and test accuracy of linear and ReLU two-layer networks on generated continuous labels on CIFAR-10. We note that for $\beta \geq 1.0$, the optimal solution for both networks is to simply set all weights to zero. The best-case test loss for the ReLU network is nearly half the best-case test loss for the linear network.

As discussed in Section 3.3., if the data matrix $X$ is spike-free, the resulting convex ReLU model (15) is similar to a linear-activation network, with the only difference being an additional cone constraint on the weight matrix $V$. It stands to wonder whether in the case of spike-free data matrices, the use of a ReLU network is necessary at all, and whether a linear-activation network would perform equally well.

In this experiment, we compare the performance of a ReLU-activation network to a linear-activation one, and demonstrate that even in the spike-free case, there exist instances in which the ReLU-activation network would be preferred. In particular, we take as our training data 3000 de-meaned and ZCA-whitened images from the CIFAR-10 dataset to form our spike-free training data $X \in \mathbb{R}^{3000 \times 3072}$. We then generate continuous labels $Y \in \mathbb{R}^{3000 \times 10}$ from a randomly-initialized ReLU two-layer network with 4000 hidden units. We use this same label-generating neural network to generate labels for images from the full 10,000-sample test set of CIFAR-10 as well, after the test images are pre-processed with the same whitening transformation used on the training data.

Across different values of $\beta$, we measured the training and generalization performance of both ReLU-activation and linear-activation two-layer neural networks trained with SGD on this dataset. Both networks used 4000 hidden units, and were trained for 400 epochs with a learning rate of $10^{-2}$ and momentum of 0.95. Our results are displayed in Figure 5.

As we can see, for all values of $\beta$, while the linear-activation network has equal or lesser training loss than the ReLU-activation network, the ReLU-activation network generalizes significantly better, achieving orders of magnitude better test loss. We should note that for values of $\beta = 1.0$ and above, both networks learn the zero network (i.e. all weights at optimum are zero), so both their training and test loss are identical to each other. We can also observe that the best-case test loss for the linear-activation network is to simply learn the zero network, whereas for a value of $\beta = 10^{-2}$ the ReLU-activation network can learn to generalize better than the zero network (achieving a test loss of 63038, compared to a test loss of 125383 of the zero-network).

These results demonstrate that even for spike-free data matrices, there are reasons to prefer a ReLU-activation network to a linear-activation network. In particular, because of the cone-

constraint on the dual weights $V$, the ReLU network is induced to learn a more complex representation than the linear network, which would explain its better generalization performance.

The CIFAR-10 dataset consists of 50,000 training images and 10,000 test images of $32 \times 32$ for 3 RGB channels, with 10 classes (Krizhevsky et al., 2009). These images were normalized by the per-channel training set mean and standard deviation. To form our training set, selected 3,000 training images from these datasets at random, where each class was equally represented. This data was then feature-wise de-meaned and transformed using ZCA. This same training class mean and ZCA transformation was also then used on the 10,000 testing points for evaluation.

### A.1.3  DOES SGD ALWAYS FIND THE GLOBAL OPTIMUM FOR NEURAL NETWORKS?

For these experiments, SGD was trained with an initial learning rate of $4 \times 10^{-5}$ for 20,000 epochs. We used a regularization penalty value of $\beta = 10^{-2}$. The value for $t$ for Algorithm 1 was found by first starting at the value of regularization penalty $\frac{1}{2} \sum_{j=1}^{m} \|u_j\|_2^2 + \|v_j\|_2^2$ from the solution from SGD, then refining this value using manual tuning. A final value of $t = 1.495$ was chosen. For this experiment, there were $P = 50$ sign patterns. Algorithm 1 was run for 30,000 iterations, and took X seconds to solve.

### A.1.4  MAXIMUM-MARGIN MATRIX FACTORIZATION

The CIFAR-10 and CIFAR-100 datasets consist of 50,000 training images and 10,000 test images of $32 \times 32$ for 3 RGB channels, with 10 and 100 classes respectively (Krizhevsky et al., 2009). These images were normalized by the per-channel training set mean and standard deviation. To form our training set, selected 3,000 training images from these datasets at random, where each class was equally represented. This data was then feature-wise de-meaned and transformed using ZCA. This same training class mean and ZCA transformation was also then used on the 10,000 testing points for evaluation. For CIFAR-10, we used a regularization parameter value of $\beta = 1.0$, whereas for CIFAR-100, we used a value of $\beta = 5.0$.

SGD was trained for 400 epochs with a learning rate of $10^{-2}$ with 1000 neurons, trained with one-hot encoded labels and squared loss. Figure 6 displays the test accuracy of the learned networks. Surprisingly the whitened classification from only 3,000 images generalizes quite well in both circumstances, far exceeding performance of the null classifier. For the CIFAR-10 experiments, the algorithm from Theorem 2 took only 0.018 seconds to solve, whereas for CIFAR-100 it took 0.36 seconds to solve.

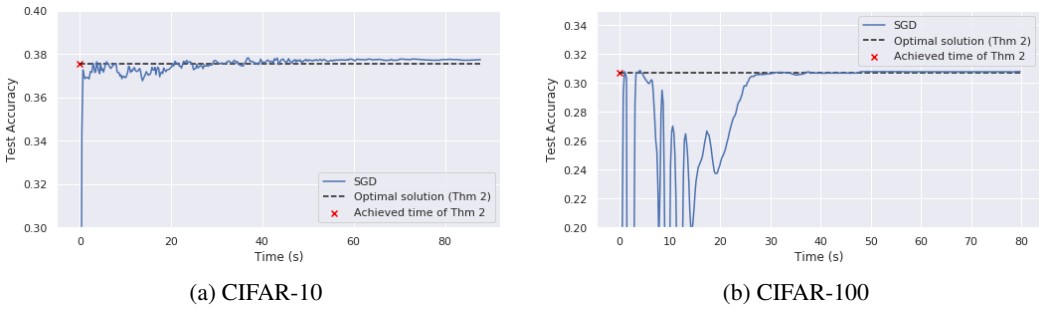

(a) CIFAR-10  (b) CIFAR-100

Figure 6: Test accuracy for maximum-margin matrix factorization and SGD for whitened CIFAR-10 and CIFAR-100.

### A.1.5  EFFECTIVENESS OF THE COPOSITIVE PROGRAM

For this classification problem, we use one-hot encoded labels and squared loss. For $\beta < 1$, SGD used a learning rate of $10^{-3}$, and otherwise used a learning rate of $2 \times 10^{-3}$. SGD was trained for 8,000 epochs with 1000 neurons, while Algorithm 1 ran for 1,000 iterations. The copositive relaxation was optimized with CVXPY with a first-order solver on a CPU with 256 GB of RAM

(Diamond & Boyd, 2016). The first-order convex solver for the copositive relaxation used a maximum of 20,000 iterations. This dataset had $P = 114$ sign patterns. The value of $t$ for Algorithm 1 was chosen as the regularization penalty $\frac{\beta}{2} \sum_{j=1}^{m} \|u_j\|_2^2 + \|v_j\|_2^2$ from the solution of SGD.

## A.2 NOTE ON DATA MATRICES OF A FIXED RANK

Consider the neural network training problem

$$\min_{u_j, v_j} \frac{1}{2} \| \sum_{j=1}^{m} (Xu_j)_+ v_j^\top - Y \|_F^2 + \frac{\beta}{2} \sum_{j=1}^{m} \|u_j\|_2^2 + \|v_j\|_2^2 \tag{21}$$

Let $X = UDV^\top$ be the compact SVD of $X$ with rank $r$, where $U \in \mathbb{R}^{n \times r}$, $D \in \mathbb{R}^{r \times r}$ and $V \in \mathbb{R}^{d \times r}$. Let $u_j' = V^\top u_j$ and $u_j^\perp = V_\perp^\top u_j$. We note that $Xu_j = XVu_j'$ and $\|u_j\|_2^2 = u_j^\top (VV^\top + V_\perp V_\perp^\top) u_j = \|u_j'\|_2^2 + \|u_j^\perp\|_2^2$. Then, we can re-parameterize the problem as

$$\min_{u_j^\perp, u_j', v_j} \frac{1}{2} \| \sum_{j=1}^{m} (XVu_j')_+ v_j^\top - Y \|_F^2 + \frac{\beta}{2} \sum_{j=1}^{m} \|u_j'\|_2^2 + \|u_j^\perp\|_2^2 + \|v_j\|_2^2 \tag{22}$$

We note that $u_j^\perp$ only appears in the regularization term. Minimizing over $u_j^\perp$ thus means simply setting it to 0. Then, we have

$$\min_{u_j', v_j} \frac{1}{2} \| \sum_{j=1}^{m} (XVu_j')_+ v_j^\top - Y \|_F^2 + \frac{\beta}{2} \sum_{j=1}^{m} \|u_j'\|_2^2 + \|v_j\|_2^2 \tag{23}$$

We note that $XV = UD \in \mathbb{R}^{n \times r}$ and $u_j' \in \mathbb{R}^r$. Thus, for $X$ of rank $r$, we can effectively reduce the dimension of the neural network training problem without loss of generality. This thus holds for all results concerning the complexity of the neural network training problem with data matrices of a fixed rank.

## A.3 PROOFS

### A.3.1 PROOF OF THEOREM 1

We begin with the primal problem (5), repeated here for convenience:

$$p^* = \min_{\substack{u_j \in \mathbb{R}^d \\ v_j \in \mathbb{R}^c}} \frac{1}{2} \| \sum_{j=1}^{m} (Xu_j)_+ v_j^\top - Y \|_F^2 + \frac{\beta}{2} \sum_{j=1}^{m} \left( \|u_j\|_2^2 + \|v_j\|_2^2 \right) \tag{24}$$

We start by re-scaling the weights in order to obtain a slightly different, equivalent objective, which has been performed previously in (Pilanci & Ergen, 2020; Savarese et al., 2019).

**Lemma 4** *The primal problem is equivalent to the following optimization problem*

$$p^* = \min_{\|u_j\|_2 \leq 1} \min_{v_j \in \mathbb{R}^c} \frac{1}{2} \| \sum_{j=1}^{m} (Xu_j)_+ v_j^\top - Y \|_F^2 + \beta \sum_{j=1}^{m} \|v_j\|_2 \tag{25}$$

**Proof:** Note that for any $\gamma > 0$, we can re-scale the parameters $\bar{u}_j = \gamma_j u_j$, $\bar{v}_j = v_j / \gamma_j$. Noting that the network output is unchanged by this re-scaling scheme, we have the equivalent problem

$$p^* = \min_{\substack{u_j \in \mathbb{R}^d \\ v_j \in \mathbb{R}^c}} \min_{\gamma_j > 0} \frac{1}{2} \| \sum_{j=1}^{m} (Xu_j)_+ v_j^\top - Y \|_F^2 + \frac{\beta}{2} \sum_{j=1}^{m} \left( \gamma_j^2 \|u_j\|_2^2 + \|v_j\|_2^2 / \gamma_j^2 \right) \tag{26}$$

Minimizing with respect to $\gamma_j$, we thus end up with

$$p^* = \min_{\substack{u_j \in \mathbb{R}^d \\ v_j \in \mathbb{R}^c}} \frac{1}{2} \| \sum_{j=1}^{m} (Xu_j)_+ v_j^\top - Y \|_F^2 + \frac{\beta}{2} \sum_{j=1}^{m} \left( \|u_j\|_2 \|v_j\|_2 \right) \tag{27}$$

We can thus set $\|u_j\|_2 = 1$ without loss of generality. Further, relaxing this constraint to $\|u_j\|_2 \leq 1$ does not change the optimal solution. In particular, for the problem

$$\min_{\|u_j\|_2 \leq 1} \min_{v_j \in \mathbb{R}^c} \frac{1}{2} \| \sum_{j=1}^{m} (Xu_j)_+ v_j^\top - Y\|_F^2 + \beta \sum_{j=1}^{m} \|v_j\|_2 \tag{28}$$

the constraint $\|u_j\|_2 = 1$ will be active for all non-zero $v_j$. Thus, relaxing the constraint will not change the objective. This proves the Lemma.

Now, we are ready to prove the first part of Theorem 1, i.e. the equivalence to the semi-infinite program (11).

**Lemma 5** *For all $\beta > 0$ primal neural network training problem (25) has a strong dual, in the form of*

$$p^* = d^* := \max_{Z: \; \|Z^\top (Xu)_+\|_2 \leq \beta \; \forall \|u\|_2 \leq 1} -\frac{1}{2} \|Z - Y\|_F^2 + \frac{1}{2} \|Y\|_F^2 \tag{29}$$

**Proof:** We first form the Lagrangian of the primal problem, by first-reparameterizing the problem as

$$\min_{\|u_j\|_2 \leq 1} \min_{v_j, R} \frac{1}{2} \|R\|_F^2 + \beta \sum_{j=1}^{m} \|v_j\|_2 \;\text{ s.t. }\; R = \sum_{j=1}^{m} (Xu_j)_+ v_j^\top - Y \tag{30}$$

and then forming the Lagrangian as

$$\min_{\|u_j\|_2 \leq 1} \min_{v_j, R} \max_{Z} \frac{1}{2} \|R\|_F^2 + \beta \sum_{j=1}^{m} \|v_j\|_2 + Z^\top Y + Z^\top R - Z^\top \Big( \sum_{j=1}^{m} (Xu_j)_+ v_j^\top \Big) \tag{31}$$

By Sion's minimax theorem, we can switch the inner maximum and minimum, and minimize over $v_j$ and $R$. This produces the following problem:

$$p^* = \min_{\|u_j\|_2 \leq 1} \max_{Z: \; \|Z^\top (Xu)_+\|_2 \leq \beta} -\frac{1}{2} \|Z - Y\|_F^2 + \frac{1}{2} \|Y\|_F^2 \tag{32}$$

We then simply need to interchange max and min to obtain the desired form. Note that this interchange does not change the objective value due to semi-infinite strong duality. In particular, for any $\beta > 0$, this problem is strictly feasible (simply let $Z = 0$) and the objective value is bounded by $\frac{1}{2} \|Y\|_F^2$. Then, by Theorem 2.2 of (Shapiro, 2009), we know that strong duality holds, and

$$p^* = d^* := \max_{Z: \; \|Z^\top (Xu)_+\|_2 \leq \beta \; \forall \|u\|_2 \leq 1} -\frac{1}{2} \|Z - Y\|_F^2 + \frac{1}{2} \|Y\|_F^2 \tag{33}$$

as desired.

Furthermore, by (Shapiro, 2009), for a signed measure $\mu$, we obtain the following strong dual of the dual program (11):

$$d^* = \max_{\mu \succeq 0} \min_{Z \in \mathbb{R}^{n \times d}} -\frac{1}{2} \|Z - Y\|_F^2 - \frac{1}{2} \|Y\|_F^2 + \int_{B_2} \Big( \|Z^\top (Xu)_+\|_2 - \beta \Big) d\mu(u) \tag{34}$$

where $B_2$ defines the unit $\ell_2$-ball. By discretization arguments in Section 3 of (Shapiro, 2009), and by Helly's theorem, there exists some $m^* \leq nc + 1$ such that this is equivalent to

$$d^* = \max_{\mu \geq 0} \min_{Z \in \mathbb{R}^{n \times d}, \|u_i\|_2 \leq 1} -\frac{1}{2} \|Z - Y\|_F^2 - \frac{1}{2} \|Y\|_F^2 + \sum_{i=1}^{m^*} \Big( \|Z^\top (Xu_i)_+\|_2 - \beta \Big) \mu_i \tag{35}$$

Minimizing with respect to $Z$, we obtain

$$\min_{\|u_i\|_2 \leq 1} \min_{\mu \geq 0} \min_{\|g_i\|_2 \leq 1} \frac{1}{2} \| \sum_{i=1}^{m^*} \mu_i (Xu_i)_+ g_i^\top - Y\|_F^2 + \beta \sum_{i=1}^{m^*} \mu_i \tag{36}$$

which we can minimize with respect to $\mu$ to obtain the finite parameterization

$$d^* = \min_{\boldsymbol{u}_i:\, \|\boldsymbol{u}_i\|_2 \leq 1} \; \min_{\boldsymbol{g}_i} \frac{1}{2} \| \sum_{i=1}^{m^*} (\boldsymbol{X}\boldsymbol{u}_i)_+ \boldsymbol{g}_i^\top - \boldsymbol{Y} \|_F^2 + \beta \sum_{i=1}^{m^*} \|\boldsymbol{g}_i\|_2 \tag{37}$$

This proves that the semi-infinite dual provides a finite support with at most $m^* \leq nc + 1$ non-zero neurons. Thus, if the number of neurons of the primal problem $m \geq m^*$, strong duality holds. Now, we seek to show the second part of Theorem 1, namely the equivalence to (12). Starting from (11), we have that the dual constraint is given by

$$\max_{\|\boldsymbol{u}\|_2 \leq 1} \|\boldsymbol{Z}^\top (\boldsymbol{X}\boldsymbol{u})_+\|_2 \leq \beta \tag{38}$$

Using the concept of dual norm, we can introduce variable $\boldsymbol{g}$ to further re-express this constraint as

$$\max_{\substack{\|\boldsymbol{u}\|_2 \leq 1 \\ \|\boldsymbol{g}\|_2 \leq 1}} \boldsymbol{g}^\top \boldsymbol{Z}^\top (\boldsymbol{X}\boldsymbol{u})_+ \leq \beta \tag{39}$$

Then, enumerating over sign patterns $\{\boldsymbol{D}_i\|_{i=1}^P$, we have

$$\max_{i \in [P]} \; \max_{\substack{\|\boldsymbol{u}\|_2 \leq 1 \\ \|\boldsymbol{g}\|_2 \leq 1 \\ (2\boldsymbol{D}_i - \boldsymbol{I})\boldsymbol{X}\boldsymbol{u} \geq 0}} \boldsymbol{g}^\top \boldsymbol{Z}^\top \boldsymbol{D}_i \boldsymbol{X}\boldsymbol{u} \leq \beta \tag{40}$$

Now, we express this in terms of an inner product.

$$\max_{\substack{i \in [P] \\ (2\boldsymbol{D}_i - \boldsymbol{I})\boldsymbol{X}\boldsymbol{u} \geq 0 \\ \|\boldsymbol{u}\|_2 \leq 1 \\ \|\boldsymbol{g}\|_2 \leq 1}} \langle \boldsymbol{Z}, \boldsymbol{D}_i \boldsymbol{X}\boldsymbol{u}\boldsymbol{g}^\top \rangle \leq \beta \tag{41}$$

Letting $\boldsymbol{V} = \boldsymbol{u}\boldsymbol{g}^\top$:

$$\max_{\substack{i \in [P] \\ \boldsymbol{V} = \boldsymbol{u}\boldsymbol{g}^\top \\ (2\boldsymbol{D}_i - \boldsymbol{I})\boldsymbol{X}\boldsymbol{u} \geq 0 \\ \|\boldsymbol{V}\|_* \leq 1}} \langle \boldsymbol{Z}, \boldsymbol{D}_i \boldsymbol{X}\boldsymbol{V} \rangle \leq \beta \tag{42}$$

Now, we can take the convex hull of the constraint set, noting that since the objective is affine, this does not change the objective value.

$$\max_{\substack{i \in [P] \\ \boldsymbol{V} \in \mathcal{K}_i \\ \|\boldsymbol{V}\|_* \leq 1}} \langle \boldsymbol{Z}, \boldsymbol{D}_i \boldsymbol{X}\boldsymbol{V} \rangle \leq \beta \tag{43}$$

Thus, the dual problem is given by

$$p^* = d^* = \max_{\boldsymbol{Z}} -\frac{1}{2} \|\boldsymbol{Z} - \boldsymbol{Y}\|_F^2 + \frac{1}{2} \|\boldsymbol{Y}\|_F^2$$
$$\text{s.t.} \max_{\substack{\boldsymbol{V}_i \in \mathcal{K}_i \\ \|\boldsymbol{V}_i\|_* \leq 1}} \langle \boldsymbol{Z}, \boldsymbol{D}_i \boldsymbol{X}\boldsymbol{V} \rangle \leq \beta \; \forall i \in [P] \tag{44}$$

We now form the Lagrangian,

$$p^* = d^* = \max_{\boldsymbol{Z}} \min_{\lambda \geq 0} \min_{\substack{\boldsymbol{V}_i \in \mathcal{K}_i \; \forall i \in [P] \\ \|\boldsymbol{V}_i\|_* \leq 1}} -\frac{1}{2} \|\boldsymbol{Z} - \boldsymbol{Y}\|_F^2 + \frac{1}{2} \|\boldsymbol{Y}\|_F^2 + \sum_{i=1}^P \lambda_i \Big( \beta - \langle \boldsymbol{Z}, \boldsymbol{D}_i \boldsymbol{X}\boldsymbol{V}_i \rangle \Big) \tag{45}$$

We note that by Sion's minimax theorem, we can switch the max and min, and then minimize over $\boldsymbol{Z}$. Following this, we obtain

$$p^* = d^* = \min_{\substack{\boldsymbol{V}_i \in \mathcal{K}_i \; \forall i \in [P] \\ \|\boldsymbol{V}_i\|_* \leq 1}} \min_{\lambda \geq 0} \frac{1}{2} \| \sum_{i=1}^P \lambda_i \boldsymbol{D}_i \boldsymbol{X}\boldsymbol{V}_i - \boldsymbol{Y} \|_F^2 + \beta \sum_{i=1}^P \lambda_i \tag{46}$$

We can re-scale our variables to obtain

$$p^* = d^* = \min_{\substack{\mathbf{V}_i \in \mathcal{K}_i \ \forall i \in [P] \\ \|\mathbf{V}_i\|_* \leq \lambda}} \min_{\lambda \geq 0} \frac{1}{2} \| \sum_{i=1}^P \mathbf{D}_i \mathbf{X} \mathbf{V}_i - \mathbf{Y} \|_F^2 + \beta \sum_{i=1}^P \lambda_i \tag{47}$$

And now minimize over $\lambda$ to obtain the desired result:

$$p^* = d^* = \min_{\mathbf{V}_i \in \mathcal{K}_i \ \forall i \in [P]} \frac{1}{2} \| \sum_{i=1}^P \mathbf{D}_i \mathbf{X} \mathbf{V}_i - \mathbf{Y} \|_F^2 + \beta \sum_{i=1}^P \|\mathbf{V}_i\|_* \tag{48}$$

**Remark 5.1** *Given the optimal solution (12), it is natural to wonder how these dual variables relate to the optimal neurons of the neural network training problem (5). Given an optimal $\{\mathbf{V}_i^*\}_{i=1}^P$, we simply need to factor them into $V_i^* = \sum_{j=1}^c \mathbf{h}_{ij}^* \mathbf{g}_{ij}^{*\top}$, where $(2\mathbf{D}_i - \mathbf{I})\mathbf{X}\mathbf{h}_{ij}^* \geq 0$. This is similar in flavor to semi-NMF. Since $\mathbf{V}_i^*$ is in the cone $\mathcal{K}_i$, exact semi-NMF factorization can be performed in polynomial time (Gillis & Kumar, 2015). Once this factorization is obtained, assuming without loss of generality that $\|\mathbf{g}_{ij}^*\|_2 = 1$, the optimal neurons are given by*

$$(\mathbf{u}_{ij}^*, \mathbf{v}_{ij}^*) = \left( \frac{\mathbf{h}_{ij}^*}{\sqrt{\|\mathbf{h}_{ij}^*\|_2}}, \mathbf{g}_{ij}^* \sqrt{\|\mathbf{h}_{ij}^*\|_2} \right), \ i \in [P], j \in [c] \tag{49}$$

*Thus, given a solution to (12), a polynomial-time algorithm exists for reconstructing weights of the original neural network training algorithm.*

### A.3.2 A COROLLARY FOR CNNS WITH AVERAGE POOLING

We first introduce additional notation for CNNs with average pooling. In particular, following Ergen & Pilanci (2020d), we use the set of patch matrices $\{\mathbf{X}_k\}_{k=1}^K$ to define a new data matrix as $\mathbf{M} := [\mathbf{X}_1, \mathbf{X}_2, \cdots \mathbf{X}_K]^\top \in \mathbb{R}^{nK \times h}$, where $h$ is the convolutional filter size. This matrix thus has a set of sign patterns, which we define as

$$P_{conv} \leq 2r_c \left( \frac{e(nK-1)}{r_c} \right)^{r_c}$$

where $r_c = \mathbf{rank}(\mathbf{M}) \leq h$. We then can enumerate the set of patch matrices

$$\{\mathbf{D}_i^k : i \in [P_{conv}], \ k \in [K]\}$$

**Corollary 5.1** *In the case of a two-layer CNN with global average pooling as in (6), the strong dual of the neural network training problem*

$$p_{conv}^* = \min_{\mathbf{V}_i \in \mathcal{K}_i \ \forall i \in [P]} \frac{1}{2} \| \sum_{i=1}^P \sum_{k=1}^K \mathbf{D}_i^k \mathbf{X}_k \mathbf{V}_i - \mathbf{Y} \|_F^2 + \beta \sum_{i=1}^P \|\mathbf{V}_i\|_* \tag{50}$$

*for convex sets $\mathcal{K}_i$, given by*

$$\mathcal{K}_i := \mathbf{conv}\{\mathbf{u}\mathbf{g}^\top : \ (2\mathbf{D}_i^k - \mathbf{I})\mathbf{X}_k\mathbf{u} \geq 0 \ \forall k, \ \|\mathbf{g}\|_2 \leq 1\} \tag{51}$$

This strong dual is convex. For a fixed kernel-size, $K$ and $r_c$ are fixed, and the problem is polynomial in $n$, i.e. of complexity proportional $\mathcal{O}(n^{r_c})$. Since in practice, $\mathbf{M}$ is a tall matrix with relatively few columns, it is almost always full-rank, in which case the computational complexity of solving the strong dual is $\mathcal{O}(n^h)$. This problem can be solved with the same Frank-Wolfe algorithm as presented in Algorithm 1.

### A.3.3 PROOF OF THEOREM 2

We note that whitened data matrices such that $n \leq d$ satisfy $\mathbf{X}\mathbf{X}^\top = \mathbf{I}$. Then, we can solve the modified problem

$$\min_{\mathbf{V} \in \mathbf{conv}\{\mathbf{u}\mathbf{g}^\top : \mathbf{X}\mathbf{u} \geq 0, \|\mathbf{g}\|_2 \leq 1\}} \frac{1}{2} \| \mathbf{V} - \mathbf{X}^\top \mathbf{Y} \|_F^2 + \beta \|\mathbf{V}\|_* \tag{52}$$

This is simply by noting that left-multiplying by $\boldsymbol{X}^\top$ does not change the norm. Now, note that

$$\min_{\boldsymbol{V}\in\mathbf{conv}\{\boldsymbol{ug}^\top:\boldsymbol{Xu}\geq 0,\|\boldsymbol{g}\|_2\leq 1\}} \frac{1}{2}\|\boldsymbol{V}-\boldsymbol{X}^\top\boldsymbol{Y}\|_F^2+\beta\|\boldsymbol{V}\|_* \geq \min_{\boldsymbol{V}\in\mathbf{conv}\{\boldsymbol{ug}^\top:\|\boldsymbol{g}\|_2\leq 1\}} \frac{1}{2}\|\boldsymbol{V}-\boldsymbol{X}^\top\boldsymbol{Y}\|_F^2+\beta\|\boldsymbol{V}\|_*$$

The relaxed problem on the right-hand side is given exactly by maximum-margin matrix factorization (Srebro et al., 2005), i.e. can be expressed as

$$\min_{\boldsymbol{V}=\sum_{i=1}^c \boldsymbol{u}_i\boldsymbol{g}_i^\top} \frac{1}{2}\|\boldsymbol{V}-\boldsymbol{X}^\top\boldsymbol{Y}\|_F^2 + \beta\|\boldsymbol{V}\|_*$$

Furthermore, this has a closed-form solution, given by soft-thresholding the singular values of $\boldsymbol{X}^\top\boldsymbol{Y}$ (Hastie et al., 2015). Thus, we have the solution of the relaxed problem as

$$\boldsymbol{V}^* = \sum_{i=1}^c (\sigma_i - \beta)_+\boldsymbol{a}_i\boldsymbol{b}_i^\top$$

Noting that $\boldsymbol{X}\boldsymbol{a}_i \geq 0$ for all $i \in \{i : \sigma_i > \beta_i\}$ by assumption, this solution is feasible for the original problem. Thus, because the optimal value obtained by the solution to the relaxed problem was a lower bound to (15), it must be the optimal solution to (15).

**Remark 5.2** *It is interesting to note that when $\boldsymbol{Y}$ is one-hot encoded,*

$$\boldsymbol{X}^\top\boldsymbol{Y} = \begin{bmatrix} n_{(1)}\mu_{(1)} & n_{(2)}\mu_{(2)} & \cdots & n_{(c)}\mu_{(c)} \end{bmatrix} \tag{53}$$

*where $n_{(i)}$ refers to the number of instances in class $i$, and $\mu_{(i)}$ refers to the mean of all data points belonging to class $i$. In this scenario, then, the optimal neural network weights, $V^*$, are found via the maximum-margin factorization of this matrix.*

*Further, if $\boldsymbol{YY}^\top$ is element-wise non-negative, by Perron-Frobenius theorem, its maximal left singular is non-negative. For such data matrices, if $\beta$ is chosen to be larger than the second largest singular value of $\boldsymbol{Y}$, the solution in (16) is guaranteed to be exact.*

### A.3.4 PROOF OF THEOREM 3

Start with (11):

$$p^* = d^* = \max_{\boldsymbol{Z}} -\frac{1}{2}\|\boldsymbol{Z} - \boldsymbol{Y}\|_F^2 + \frac{1}{2}\|\boldsymbol{Y}\|_F^2$$
$$\text{s.t } \|\boldsymbol{Z}^\top(\boldsymbol{Xu})_+\|_2 \leq \beta \ \forall \|\boldsymbol{u}\|_2 \leq 1 \tag{54}$$

We can express this as

$$p^* = d^* = \max_{\boldsymbol{Z}} -\frac{1}{2}\|\boldsymbol{Z} - \boldsymbol{Y}\|_F^2 + \frac{1}{2}\|\boldsymbol{Y}\|_F^2$$
$$\text{s.t } \max_{\|\boldsymbol{u}\|_2\leq 1} \|\boldsymbol{Z}^\top(\boldsymbol{Xu})_+\|_2^2 \leq \beta^2 \tag{55}$$

Enumerating over all possible sign patterns $i \in [P]$, and noting that $\|\boldsymbol{Z}^\top\boldsymbol{D}_i\boldsymbol{Xu}\|_2^2 = \mathbf{tr}\Big(\boldsymbol{Z}^\top\boldsymbol{D}_i\boldsymbol{Xuu}^\top\boldsymbol{X}^\top\boldsymbol{D}_i\boldsymbol{Z}\Big)$, we have

$$p^* = d^* = \max_{\boldsymbol{Z}} -\frac{1}{2}\|\boldsymbol{Z} - \boldsymbol{Y}\|_F^2 + \frac{1}{2}\|\boldsymbol{Y}\|_F^2$$
$$\text{s.t } \max_{\substack{i\in[P]\\(2\boldsymbol{D}_i-\boldsymbol{I})\boldsymbol{Xu}\geq 0\\\|\boldsymbol{u}\|_2\leq 1}} \mathbf{tr}\Big(\boldsymbol{Z}^\top\boldsymbol{D}_i\boldsymbol{Xuu}^\top\boldsymbol{X}^\top\boldsymbol{D}_i\boldsymbol{Z}\Big) \leq \beta^2 \tag{56}$$

Noting the maximization constraint is linear in $\boldsymbol{uu}^\top$, we can take the convex hull of the constraint set and not change the optimal value. Thus,

$$p^* = d^* = \max_{\boldsymbol{Z}} -\frac{1}{2}\|\boldsymbol{Z} - \boldsymbol{Y}\|_F^2 + \frac{1}{2}\|\boldsymbol{Y}\|_F^2$$
$$\text{s.t } \max_{\substack{i\in[P]\\\boldsymbol{U}\in\mathcal{C}_i\\\mathbf{tr}(\boldsymbol{U})\leq 1}} \mathbf{tr}\Big(\boldsymbol{Z}^\top\boldsymbol{D}_i\boldsymbol{XUX}^\top\boldsymbol{D}_i\boldsymbol{Z}\Big) \leq \beta^2 \tag{57}$$

Which is then equivalent to

$$p^* = d^* = \max_{\boldsymbol{Z}} -\frac{1}{2}\|\boldsymbol{Z} - \boldsymbol{Y}\|_F^2 + \frac{1}{2}\|\boldsymbol{Y}\|_F^2$$

$$\text{s.t} \max_{\substack{\boldsymbol{U}_i \in \mathcal{C}_i \\ \mathbf{tr}(\boldsymbol{U}_i) \le 1}} \mathbf{tr}\left(\boldsymbol{Z}^\top \boldsymbol{D}_i \boldsymbol{X} \boldsymbol{U}_i \boldsymbol{X}^\top \boldsymbol{D}_i \boldsymbol{Z}\right) \le \beta^2 \ \forall i \in [P] \tag{58}$$

We thus have a problem with a convex objective and $P$ constraints, so we can take the Lagrangian

$$p^* = d^* = \max_{\boldsymbol{Z}} \min_{\substack{\boldsymbol{U}_i \in \mathcal{C}_i \ \forall i \in [P] \\ \mathbf{tr}(\boldsymbol{U}_i) \le 1 \\ \lambda \ge 0}} -\frac{1}{2}\|\boldsymbol{Z} - \boldsymbol{Y}\|_F^2 + \frac{1}{2}\|\boldsymbol{Y}\|_F^2 + \sum_{i=1}^P \lambda_i \left(\beta^2 - \mathbf{tr}\left(\boldsymbol{Z}^\top \boldsymbol{D}_i \boldsymbol{X} \boldsymbol{U}_i \boldsymbol{X}^\top \boldsymbol{D}_i \boldsymbol{Z}\right)\right) \tag{59}$$

Noting that this function is convex over $\boldsymbol{Z}$, affine over $\lambda$, and concave over $\boldsymbol{U}_i$, and the constraint set is convex, by Sion's minimax theorem we can change max and min without changing the objective. Thus,

$$p^* = d^* = \min_{\substack{\boldsymbol{U}_i \in \mathcal{C}_i \ \forall i \in [P] \\ \mathbf{tr}(\boldsymbol{U}_i) \le 1 \\ \lambda \ge 0}} \max_{\boldsymbol{Z}} -\frac{1}{2}\|\boldsymbol{Z} - \boldsymbol{Y}\|_F^2 + \frac{1}{2}\|\boldsymbol{Y}\|_F^2 + \sum_{i=1}^P \lambda_i \left(\beta^2 - \mathbf{tr}\left(\boldsymbol{Z}^\top \boldsymbol{D}_i \boldsymbol{X} \boldsymbol{U}_i \boldsymbol{X}^\top \boldsymbol{D}_i \boldsymbol{Z}\right)\right) \tag{60}$$

Solving for $\boldsymbol{Z}$ and re-substituting, we thus have

$$p^* = d^* = \min_{\substack{\boldsymbol{U}_i \in \mathcal{C}_i \\ \mathbf{tr}(\boldsymbol{U}_i) \le 1 \\ \lambda \ge 0}} \frac{1}{2}\mathbf{tr}\left(\boldsymbol{Y}^\top \left(\boldsymbol{I} + 2\sum_{i=1}^P \lambda_i (\boldsymbol{D}_i \boldsymbol{X}) \boldsymbol{U}_i (\boldsymbol{D}_i \boldsymbol{X})^\top\right)^{-1} \boldsymbol{Y}\right) + \beta^2 \sum_{i=1}^P \lambda_i \tag{61}$$

as in the proof of Theorem 1, we easily re-scale this to obtain the desired objective:

$$p^* = d^* = \min_{\boldsymbol{U}_i \in \mathcal{C}_i \ \forall i \in [P]} \frac{1}{2}\mathbf{tr}\left(\boldsymbol{Y}^\top \left(\boldsymbol{I} + 2\sum_{i=1}^P (\boldsymbol{D}_i \boldsymbol{X}) \boldsymbol{U}_i (\boldsymbol{D}_i \boldsymbol{X})^\top\right)^{-1} \boldsymbol{Y}\right) + \beta^2 \sum_{i=1}^P \mathbf{tr}(\boldsymbol{U}_i) \tag{62}$$

### A.4 Notes on the Frank-Wolfe Method Applied to (12)

#### A.4.1 Derivation of the Frank-Wolfe Algorithm

In general, the Frank-Wolfe algorithm (Frank et al., 1956) aims to solve the problem

$$\min_{\boldsymbol{x}} f(\boldsymbol{x}) \text{ s.t. } \boldsymbol{x} \in \mathcal{D} \tag{63}$$

for some convex function $f$ and convex set $\mathcal{D}$. It does so in the following iterative steps $k = 1, \cdots, K$ with step sizes $\alpha^{(k)}$ and an initial point $\boldsymbol{x}^{(1)}$

1. (LMO step) Solve the problem

$$\boldsymbol{m}^{(k)} := \arg\min_{\boldsymbol{s} \in \mathcal{D}} \langle \boldsymbol{s}, \nabla_{\boldsymbol{x}} f(\boldsymbol{x}^{(k)}) \rangle \tag{64}$$

2. (Update step) Update the decision variable

$$\boldsymbol{x}^{(k+1)} = (1 - \alpha^{(k)})\boldsymbol{x}^{(k)} + \alpha^{(k)}\boldsymbol{m}^{(k)} \tag{65}$$

We will now apply the Frank-Wolfe problem to the inner minimization of (14), i.e., for a fixed value of $t$, solving

$$\min_{\substack{\boldsymbol{V}_i \in \mathcal{K}_i \ \forall i \in [P] \\ \sum_{i=1}^P \|\boldsymbol{V}_i\|_* \le t}} \frac{1}{2}\|\sum_{i=1}^P \boldsymbol{D}_i \boldsymbol{X} \boldsymbol{V}_i - \boldsymbol{Y}\|_F^2 + \beta t \tag{66}$$

In particular the LMO step becomes:

$$\{\boldsymbol{M}_i^{(k)}\}_{i=1}^P := \arg \min_{\substack{\boldsymbol{S}_i \in \mathcal{K}_i \; \forall i \in [P] \\ \sum_{i=1}^P \|\boldsymbol{S}_i\|_* \leq t}} \sum_{i=1}^P \left\langle \boldsymbol{S}_i, (\boldsymbol{D}_i \boldsymbol{X})^\top \Big( \sum_{j=1}^P \boldsymbol{D}_j \boldsymbol{X} \boldsymbol{V}_j^{(k)} - \boldsymbol{Y} \Big) \right\rangle \tag{67}$$

$$= \arg \min_{\substack{\boldsymbol{S}_i \in \mathcal{K}_i \; \forall i \in [P] \\ \sum_{i=1}^P \|\boldsymbol{S}_i\|_* \leq t}} \sum_{i=1}^P \left\langle \boldsymbol{D}_i \boldsymbol{X} \boldsymbol{S}_i, \sum_{j=1}^P \boldsymbol{D}_j \boldsymbol{X} \boldsymbol{V}_j^{(k)} - \boldsymbol{Y} \right\rangle \tag{68}$$

Note that the objective value in the optimization problem of the LMO step is affine with respect to the variables $\boldsymbol{S}_i$. Thus, the optimal value occurs at an extreme point of the feasible set. Thus, only one of $\{\boldsymbol{M}_i\}_{i=1}^P$ is active at optimum at index $i = i^*$, and moreover this $\boldsymbol{M}_{i^*}$ occurs at an extreme point of $\mathcal{K}_{i^*}$, i.e $\boldsymbol{M}_{i^*} \in \bar{\mathcal{K}}_{i^*}$. All other $\boldsymbol{M}_i$ where $i \neq i^*$ are zero at optimum. Thus, we can re-write this problem as

$$\boldsymbol{M}_{i^*} = \arg \min_{\substack{i \in [P] \\ \boldsymbol{S} \in \bar{\mathcal{K}}_i \\ \|\boldsymbol{S}\|_* \leq t}} \left\langle \boldsymbol{D}_i \boldsymbol{X} \boldsymbol{S}, \sum_{j=1}^P \boldsymbol{D}_j \boldsymbol{X} \boldsymbol{V}_j^{(k)} - \boldsymbol{Y} \right\rangle \tag{69}$$

Recalling that $\mathcal{K}_i := \mathbf{conv}\{\boldsymbol{u}\boldsymbol{g}^\top : (2\boldsymbol{D}_i - \boldsymbol{I})\boldsymbol{X}\boldsymbol{u} \geq 0, \|\boldsymbol{g}\|_2 \leq 1\}$ boundary of $\mathcal{K}_i$ is simple to express, leaving us with $\boldsymbol{M}_{i^*} = \boldsymbol{u}_{i^*}\boldsymbol{g}_{i^*}^\top$, where

$$(\boldsymbol{u}_{i^*}, \boldsymbol{g}_{i^*}) = \arg \min_{\substack{i \in [P] \\ (2\boldsymbol{D}_i - \boldsymbol{I})\boldsymbol{X}\boldsymbol{u} \geq 0 \\ \|\boldsymbol{g}\|_2 \leq 1 \\ \|\boldsymbol{u}\|_2 \leq t}} \left\langle \boldsymbol{D}_i \boldsymbol{X} \boldsymbol{u} \boldsymbol{g}^\top, \sum_{j=1}^P \boldsymbol{D}_j \boldsymbol{X} \boldsymbol{V}_j^{(k)} - \boldsymbol{Y} \right\rangle \tag{70}$$

Note that we can change the constraint $\|\boldsymbol{u}\|_2 \leq t$ to $\|\boldsymbol{u}\|_2 \leq 1$ and simply multiply by a factor of $t$ to the solution afterwards. We can thus write the key Frank-Wolfe LMO step subproblem as

$$(\boldsymbol{u}_{i^*}, \boldsymbol{g}_{i^*}) = \max_{\substack{i \in [P] \\ (2\boldsymbol{D}_i - \boldsymbol{I})\boldsymbol{X}\boldsymbol{u} \geq 0 \\ \|\boldsymbol{g}\|_2 \leq 1 \\ \|\boldsymbol{u}\|_2 \leq 1}} \langle \boldsymbol{D}_i \boldsymbol{X} \boldsymbol{u} \boldsymbol{g}^\top, \boldsymbol{Y} - \sum_{j=1}^P \boldsymbol{D}_j \boldsymbol{X} \boldsymbol{V}_j^{(k)} \rangle \tag{71}$$

as desired. For each $i \in [P]$, we thus solve the problem

$$s_i^{(k)} = \max_{\substack{(2\boldsymbol{D}_i - \boldsymbol{I})\boldsymbol{X}\boldsymbol{u} \geq 0 \\ \|\boldsymbol{g}\|_2 \leq 1 \\ \|\boldsymbol{u}\|_2 \leq 1}} \langle \boldsymbol{D}_i \boldsymbol{X} \boldsymbol{u} \boldsymbol{g}^\top, \boldsymbol{Y} - \sum_{j=1}^P \boldsymbol{D}_j \boldsymbol{X} \boldsymbol{V}_j^{(k)} \rangle \tag{72}$$

and store the arg maxes $(\boldsymbol{u}_i, \boldsymbol{g}_i)$ for each $i$. Then, from the index which attains the maximum $i^* := \arg\max_i s_i^{(k)}$, form $\boldsymbol{M}_{i^*}^{(k)} = \boldsymbol{u}_{i^*}\boldsymbol{g}_{i^*}^\top$. For all other $i \neq i^*$, as stated previously, $\boldsymbol{M}_i^{(k)} = \boldsymbol{0}$. Then, we must re-multiply by the factor $t$ which was removed in (71) to obtain the update rule for all $i \in [P]$:

$$\boldsymbol{V}_i^{(k+1)} = (1 - \alpha^{(k)})\boldsymbol{V}_i^{(k)} + t\alpha^{(k)}\boldsymbol{M}_i^{(k)}$$

### A.4.2 SOLVING EACH FRANK-WOLFE ITERATE

We discuss here how to solve the subproblem

$$s_i^{(k)} = \max_{\substack{\|\boldsymbol{u}\|_2 \leq 1 \\ \|\boldsymbol{g}\|_2 \leq 1 \\ (2\boldsymbol{D}_i - \boldsymbol{I})\boldsymbol{X}\boldsymbol{u} \geq 0}} \langle \boldsymbol{D}_i \boldsymbol{X} \boldsymbol{u} \boldsymbol{g}^\top, \boldsymbol{Y} - \sum_{j=1}^P \boldsymbol{D}_i \boldsymbol{X} \boldsymbol{V}_i^{(k)} \rangle$$

using cone-constrained PCA. In particular, note that for a fixed $\boldsymbol{u}$, we can solve for $\boldsymbol{g}$ in closed form. Let $\boldsymbol{R}^{(k)} = \boldsymbol{Y} - \sum_{j=1}^{P} \boldsymbol{D}_i \boldsymbol{X} \boldsymbol{V}_i^{(k)}$. Then, we have

$$\boldsymbol{g}^*(\boldsymbol{u}) = \frac{\boldsymbol{R}^{(k)\top} \boldsymbol{D}_i \boldsymbol{X} \boldsymbol{u}}{\|\boldsymbol{R}^{(k)\top} \boldsymbol{D}_i \boldsymbol{X} \boldsymbol{u}\|_2}$$

Then, re-substituting this back into the objective, we have

$$s_i^{(k)} = \max_{\substack{\|\boldsymbol{u}\|_2 \leq 1 \\ (2\boldsymbol{D}_i - \boldsymbol{I})\boldsymbol{X}\boldsymbol{u} \geq 0}} \|\boldsymbol{R}^{(k)\top} \boldsymbol{D}_i \boldsymbol{X} \boldsymbol{u}\|_2 = \max_{\substack{\|\boldsymbol{u}\|_2 \leq 1 \\ (2\boldsymbol{D}_i - \boldsymbol{I})\boldsymbol{X}\boldsymbol{u} \geq 0}} \sqrt{\boldsymbol{u}^\top (\boldsymbol{D}_i \boldsymbol{X})^\top \boldsymbol{R}^{(k)} \boldsymbol{R}^{(k)\top} (\boldsymbol{D}_i \boldsymbol{X}) \boldsymbol{u}}$$

Without loss of generality, we can square the objective, since objective only takes on non-negative values. The LMO step of the Frank-Wolfe algorithm is thus identical to the cone-constrained PCA problem

$$s_i^{(k)2} = \max_{\substack{\|\boldsymbol{u}\|_2 \leq 1 \\ (2\boldsymbol{D}_i - \boldsymbol{I})\boldsymbol{X}\boldsymbol{u} \geq 0}} \boldsymbol{u}^\top (\boldsymbol{D}_i \boldsymbol{X})^\top \boldsymbol{R}^{(k)} \boldsymbol{R}^{(k)\top} (\boldsymbol{D}_i \boldsymbol{X}) \boldsymbol{u}$$

This problem, via Lemma 7 of Asteris et al. (2014), can be performed in $\mathcal{O}(n^d)$ time. However, we note that because we can reduce the dimension of the problem to $r$ without loss of generality (see Appendix A.2), this thus simplifies to $\mathcal{O}(n^r)$. We perform $P$ of such maximization problems per iteration of Frank-Wolfe.

### A.5 A CUTTING PLANE METHOD FOR SOLVING (17)

Consider the dual problem

$$d^* = \max_{\boldsymbol{Z}} -\frac{1}{2}\|\boldsymbol{Z} - \boldsymbol{Y}\|_F^2 + \frac{1}{2}\|\boldsymbol{Y}\|_F^2$$
$$\text{s.t} \max_{\|\boldsymbol{u}\|_2 \leq 1} \|\boldsymbol{Z}^\top (\boldsymbol{X}\boldsymbol{u})_+\|_2^2 \leq \beta^2 \tag{73}$$

Enumerating over sets of hyperplanes, we have

$$p^* = d^* = \max_{\boldsymbol{Z}} -\frac{1}{2}\|\boldsymbol{Z} - \boldsymbol{Y}\|_F^2 + \frac{1}{2}\|\boldsymbol{Y}\|_F^2$$
$$\text{s.t} \max_{\substack{i \in [P] \\ (2\boldsymbol{D}_i - \boldsymbol{I})\boldsymbol{X}\boldsymbol{u} \geq 0 \\ \|\boldsymbol{u}\|_2 \leq 1}} \|\boldsymbol{Z}^\top \boldsymbol{D}_i \boldsymbol{X} \boldsymbol{u}\|_2^2 \leq \beta^2 \tag{74}$$

We can express the dual constraint as

$$\max_{\substack{i \in [P] \\ (2\boldsymbol{D}_i - \boldsymbol{I})\boldsymbol{X}\boldsymbol{u} \geq 0 \\ \|\boldsymbol{u}\|_2 \leq 1}} \boldsymbol{u}^\top \boldsymbol{M}_i \boldsymbol{u} \leq \beta^2 \tag{75}$$

where $\boldsymbol{M}_i = \boldsymbol{X}^\top \boldsymbol{D}_i \boldsymbol{Z} \boldsymbol{Z}^\top \boldsymbol{D}_i \boldsymbol{X} \in \mathbb{R}^{d \times d}$. For each $i$, this is a cone-constrained PCA problem in dimension $d$ with $n$ linear inequality constraints. Using Lemma 7 of (Asteris et al., 2014), there exists an algorithm which solves the cone-constrained PCA problem in $\mathcal{O}(n^r)$ time. Thus, solving the optimization problem in the dual constraint (75) is $\mathcal{O}(Pn^r)$ time. For a fixed rank $r$, $P$ has order $\mathcal{O}(n^r)$, but otherwise $P$ is $\mathcal{O}(n^d)$ as well. Thus, determining the feasibility of a dual variable $Z$ from (75) has complexity $\mathcal{O}(n^d)$ in the general case, and $\mathcal{O}(n^r)$ in the rank-$r$ case.

Now, using the Analytic Center Cutting Plane Method (ACCPM), the cone-constrained PCA procedure above provides an oracle for the feasibility of a dual variable $\boldsymbol{Z}$. Using this oracle, ACCPM solves the convex problem (73) in polynomial steps in terms of the dimension of $\boldsymbol{Z}$, i.e. $\textbf{poly}(nc)$ steps. Thus, the overall complexity of the cutting plane method is $\mathcal{O}(n^d \textbf{poly}(nc))$ in the general case, and $\mathcal{O}(n^r \textbf{poly}(nc))$ in the rank-$r$ case, which is polynomial for a fixed $r$.

We note that while this is similar in complexity to the results of (Pilanci & Ergen, 2020), there are a few subtle differences. In particular, there are additional terms polynomial in $c$ which do not appear in their work. However, the broader picture of the analyses align–for full-rank matrices, solving the convex dual of the neural network training problem is NP-hard, while for matrices of a fixed rank, a polynomial time algorithm exists.

## A.6 EXTENSIONS TO GENERAL CONVEX LOSS FUNCTIONS

This follows closely from a similar discussion by Pilanci & Ergen (2020). Consider the objective of the neural network training problem with a general convex loss function $\ell$, given by

$$p^* = \min_{\{\boldsymbol{u}_j\}_{j=1}^m, \{\boldsymbol{v}_j\}_{j=1}^m} \frac{1}{2}\ell(\sum_{j=1}^m (\boldsymbol{X}\boldsymbol{u}_j)_+ \boldsymbol{v}_j^\top, \boldsymbol{Y}) + \frac{\beta}{2}\sum_{j=1}^m \left(\|\boldsymbol{u}_j\|_2^2 + \|\boldsymbol{v}_j\|_2^2\right) \tag{76}$$

Then, we can follow the same proof of Theorem 1 exactly, instead substituting the Fenchel dual of $\ell$, defined as

$$\ell^*(\boldsymbol{Z}) = \max_{\boldsymbol{R}} \langle \boldsymbol{Z}, \boldsymbol{R} \rangle - \ell(\boldsymbol{R}, \boldsymbol{Y}) \tag{77}$$

Then, we have the dual objective analog of (11):

$$\max_{\boldsymbol{Z}:\|\boldsymbol{Z}^\top (\boldsymbol{X}\boldsymbol{u})_+\|_2 \leq \beta \ \forall \|\boldsymbol{u}\|_2 \leq 1} -\ell^*(\boldsymbol{Z}) \tag{78}$$

We can further elucidate the convex constraint on dual variable $\boldsymbol{Z}$ as done in the proof of Theorem 1. Then, we can follow the same steps from this previous proof, noting that by Fenchel–Moreau Theorem, $\ell^{**} = \ell$ (Borwein & Lewis, 2010). Thus, the finite-dimensional convex strong dual is given by

$$p^* = d^* = \min_{\boldsymbol{V}_i \in \mathcal{K}_i \ \forall i \in [P]} \frac{1}{2}\ell\Big(\sum_{i=1}^P \boldsymbol{D}_i \boldsymbol{X}\boldsymbol{V}_i, \boldsymbol{Y}\Big) + \beta \sum_{i=1}^P \|\boldsymbol{V}_i\|_* \tag{79}$$

as desired. We note that the Frank-Wolfe method in Algorithm 1 holds for general convex $\ell$ as well, with a small modification corresponding to the gradient of the loss function.

