# OpenReview forum: "Vector-output ReLU Neural Network Problems are Copositive Programs: Convex Analysis of Two Layer Networks and Polynomial-time Algorithms"
_ICLR.cc/2021/Conference — ICLR 2021 Poster_

### Official Review · AnonReviewer2 · 2020-10-19
**an interesting theory paper that shows connection between two-layer ReLU network training and convex copositive program**

**Rating:** 7
**Confidence:** 4

**Review:**

This paper showed that a two-layer vector-output ReLU neural network training problem is equivalent to a finite-dimensional convex copositive program. Based on this connection, the authors gave the first algorithm that finds the global min of the network training problem, which has running time polynomial in the number of samples but exponential in the data matrix. For CNN, the running time is only exponential in the filter size, which is usually a constant. The authors also described circumstances in which the global min can be efficiently found by soft-thresholded SVD; provided a copositive relaxation that is exact for certain cases. The effectiveness of the proposed algorithms is verified in experiments.

I think this is a good theory paper that shows an interesting connection between two layer network training and copositive program. My questions/concerns are as follows:

1. What’s the role of the regularizer in the current analysis? Does the result extend to the setting where the loss function does not have a regularizer?
2. The current analysis seems to be restricted to two-layer neural networks. I understand the analysis of multiple layer neural networks can be potentially much more difficult. It would be good if the authors can add some discussion on the possibility of extending the analysis to more general multiple-layer nets.
3. The current paper focuses on minimizing the training loss. In practice, people usually use over-parameterized neural networks which has an infinite number of global min for training loss. It’s believed that SGD works well because SGD has an implicit bias which tends to converge to a global min that also generalizes well. So I think it would be good if the authors can check the generalization performance of the proposed new algorithm.
4. Since the algorithm in this paper is limited to a two-layer ReLU neural network, which is rarely used in practice. So I wonder what’s the general implication/message of this paper to the practical ML community.
5. A closely related paper [Ge et al 2018] also studies two-layer ReLU neural networks with vector-output. Under the assumption of symmetric distribution and more output dimension than hidden neurons, they gave a tensor decomposition algorithm that can recover the ground truth parameters in polynomial time. It might be good to compare the current paper and their result.
https://arxiv.org/abs/1810.06793

---

> ### Author Response · Authors · 2020-11-16
> **Thank you for the review. (see cont.)**
>
> We thank the reviewer for their comments and suggestions. Please see our point-to-point response below.
>
> 1. "What’s the role of the regularizer in the current analysis? Does the result extend to the setting where the loss function does not have a regularizer?"
>
> The regularizer in our analysis is what allows for strong duality to occur. In particular, $\beta > 0$ is required for Slater’s condition to hold, and thereby for the step from (29) to (30) in Appendix A.3.1. In this work we only studied standard $\ell_2$ regularization on all of the weights because this is what is commonly used in the form of weight decay, but other regularizers could also be explored in future works.
>
> In the case without a regularizer, the current strong duality proof as it is would not hold. It is generally known that neural networks without regularizers have many global training minima (i.e. the training loss goes to 0), and then it is more of a question of which global minima generalize well, and how to find them. We leave this line of research to other works, such as explored by (Gunasekar et al., 2017; Neyshabur et al., 2014), which study the implicit bias of neural networks trained without regularizers. By introducing the regularizer, we do not need to worry as much about generalization properties, since well-tuned weight decay can perform as well as if not better than unregularized models.
>
> 2. "The current analysis seems to be restricted to two-layer neural networks. I understand the analysis of multiple layer neural networks can be potentially much more difficult. It would be good if the authors can add some discussion on the possibility of extending the analysis to more general multiple-layer nets."
>
> Yes, this is a good point. We do mention that greedily training two-layer networks can extend this to deeper networks, which would allow for our analysis to such settings, in the related work section. A similar method has been shown to perform as well as end-to-end trained networks such as VGG on ImageNet image classification tasks (Belilovsky et al., 2019). We have reiterated this point in the conclusion to further emphasize this point. As for training deeper networks end-to-end, it is true that the analysis becomes much more difficult. We also mention in the revised conclusion an idea for what this would look like.
>
> 3. "The current paper focuses on minimizing the training loss. In practice, people usually use over-parameterized neural networks which has an infinite number of global min for training loss. It’s believed that SGD works well because SGD has an implicit bias which tends to converge to a global min that also generalizes well. So I think it would be good if the authors can check the generalization performance of the proposed new algorithm."
>
> As we mention in the related work section of our paper, networks with well-tuned weight decay can perform as well as unregularized neural networks (Gunakesar et al., 2019). Since we can provably find the global minimum of a network with weight decay with the convex dual optimization problem, it thus stands to reason that the solution found by the dual problem would generalize well when $\beta$ is well-tuned.
>
> As an example, in the paper, we compare the generalization performance between SGD and the global minimum found by soft-thresholded SVD on classifying images from CIFAR-10 and CIFAR-100 in appendix A.1.4. As we can see in Figure 6, the two methods generalize equally as well.
>
> 4. "Since the algorithm in this paper is limited to a two-layer ReLU neural network, which is rarely used in practice. So I wonder what’s the general implication/message of this paper to the practical ML community."
>
> We believe that this is just the first step in understanding the performance of deep neural networks as they are used in practice. For practical insights, we again refer to the concept of greedily training shallow networks as from (Belilovsky et al., 2019), which can allow for these two-layer networks to in fact become very powerful (e.g. exceed the performance of AlexNet and VGG networks by sequentially training two-layer networks).

---

> > ### Author Response · Authors · 2020-11-16
> > **Thank you for the review. (cont.)**
> >
> > 5. "A closely related paper [Ge et al 2018] also studies two-layer ReLU neural networks with vector-output. Under the assumption of symmetric distribution and more output dimension than hidden neurons, they gave a tensor decomposition algorithm that can recover the ground truth parameters in polynomial time. It might be good to compare the current paper and their result."
> >
> > We thank the reviewer for bringing this paper to our attention. This is quite an interesting work and we have cited the result, along with another related result from (Brutkus & Globerson, 2017) in our related work section. As you have mentioned, their result depends on many assumptions on the data, such as an under-parameterized network (output dimension greater than number of neurons), as well as the data being generated from a planted neural network model (i.e. there is a ground truth). The model considered by this work also assumes a distribution from which data can be drawn from arbitrarily many times, rather than a fixed dataset over which optimization must occur. Thus, it is hard to exactly compare to their result.
> >
> > For general datasets, we have shown that the worst-case complexity of solving the neural network cannot be better than exponential in the dimension via reduction to non-negative PCA. Thus, the assumption on the data distribution from (Ge et al., 2018) seems to be crucial for the success of their algorithm. This is also mirrored in the work from (Brutkus & Globerson, 2017) who show that for the model considered in their work, under general input distributions finding the global optimum is NP-complete.
> >
> > **References**
> > Alon Brutzkus and Amir Globerson. Globally optimal gradient descent for a convnet with gaussian inputs. *arXiv preprint arXiv:1702.07966*, 2017.

---

### Official Review · AnonReviewer1 · 2020-10-27
**An important study advancing our understanding of the optimization of neural networks**

**Rating:** 7
**Confidence:** 5

**Review:**

Summary: This paper generalizes the results of Pilanci and Ergen (2020) showing that the non-convex optimization problem corresponding to the training of a one-hidden-layer muti-output ReLU neworks can be solved using convex programming. In particular they show that the problem has (I) a finite convex bidual that can be solved efficiently using some variant of the Frank-Wolfe algorithm and (II) a convex strong dual given by a copositive program. Unfortunately in the general case the complexity is exponential in the rank of the data matrix. A spike-free assumption is introduced that facilitates things and allows polynomial time algorithms, however this assumption is pretty strong and I doubt it is useful in practice as it makes the training 'almost' equivalent to training a linear classifier. Some references are missing as the problem is related to the low-rank matrix factorization problem. The notation in some critical parts of the paper is not clear and makes reading difficult.

I would like to start saying that I am open to increasing my score (**Update: score updated after author feedback**) if the authors are able to clarify notation and add references/discussion, because I think that the essential contributions of this paper are important

Pros:
1. Quality/Significance: The paper continues the vein of previous convex reformulations of the training of ReLU networks of Pilanci-Ergen (2020), extending the results of the single output case to multi-output which is arguably the interesting case in contemporary applications of Deep Learning. It opens the way to certified global optimality of solutions via convex programming of ReLU networks, and provides insight into the fundamental complexity of the optimization problem in nonconvex (factored) vs
convex form.

2. Originality: There are few papers trying to understand the connections between the nonconvex formulation of the training of ReLU networks with convex formulations that provide certified optimality. For this reason I find the paper original although it builds upon previous work exploring this idea. Most papers deal with heuristic or ad-hoc arguments of convergence of GD/SGD in the non-convex formulation with many assumptions that might not hold in practice or that are cumbersome.

3. Clarity: The "text" part of the paper is clearly written and the pace seems good. However there are crucial points where I could not understand the notation (see cons).

Cons:
1. **This has been fixed in an updated version and is now not a problem** Clarity: I could not really understand the optimization problem (9) and (11) because the variable $i$ appears as an optimization variable $\min_{i \in [P]}$ but it does not appear in the optimization objective? In the objective the letter $i$ is used but as an index in the summation $\sum_{i=1}^P$ which makes things really confusing. It appears that $i$ is not really an optimization variable and perhaps what the authors meant is that the constraint in (9) and (11) should read $V_i \in K_i \forall i \in [P]$? at least this is what I would find most natural. This should be clarified and corrected if needed.

2. **This has been fixed in an updated version and is now not a problem** Clarity: It is not so clear how Algorithm 1 follows the Frank-Wolfe template. As I understand an initial value of $t$ is chosen and then the inner minimization problem is solved. Following this a new value of t is chosen and so on. So in the inner minimization problem the constraint set (assuming my understanding in the previous point) is $\sum_{i=1}^P ||V_i|| \leq t$ . Then the step (a) looks like the LMO but it is not clear where is the constraint \sum_{i=1}^P \| V_i \|_ enforced? It looks like the FW update necessarily modifies only one $V_i$ and that the solution can be obtained from the LMO corresponding to only one constraint ||V_i||_ \leq 1 is this the case? I think it is enforced through the constraint $\|u\|_2 \leq 1$.
It would be great if the authors can confirm and flesh out the intermediate steps in the appendix, which does not explain much besides the algorithm in the main text. This would make it more accesible.

3. **The authors argue in rebuttal that ReLU networks on spike-free data is still different from a linear classifier, some experiments were added** Significance: The Spike-free assumption seems to simplify things but it might be unrealistic. what is even more evident is that
it is somehow equivalent to learning a linear classifier as it implies that $(XU)_+ = XU$ so the network is actually $XUV^T$ so it is equivalent to $XA$ with $A=UV^T$. Under this identification it is well known that the variational formulation of the nuclear norm
implies that the regularizer $\|U\|_F^2 + \|V\|_F^2$ is equivalent to the nuclear norm of the unfactored matrix $A$. There are multiple works studying this that should be mentioned and it should be acknowledge that the spike-free assumption is more convenient for simplifying the analysis, rather than realistic (or it should be argued why it is ok to do the assumption).  Also it is not clear in the statement of the theorem what is the relation to spike-free. Does whitened $X$ imply spike-free $X$?

4. Significance: Theorem 2 is a simple consequence of this "spike-free" simplified setting and corresponds to the solution of the proximal operator of the nuclear norm.

5. Significance: the previous two points suggest that the important results here are those corresponding to general $X$ matrix. Unfortunately in that case the complexity is exponential in the rank of the data, which I guess is just the "real" dimensionality of the data? for example if data is collinear this would be 1 and it is not surprising that the problem would become easier. It looks like the proposed algorithms are currently only of theoretical interest.

6. **Authors have answered this in the revision** Experiments: exp 5.1: Perhaps add here the corresponding plots for the 0-1 error? it would be great to understand if the solution obtained with convex programming translates to better misclassification error compared to sgd. Currently how much time does the convex program take to solve? compared to SGD
do you think there is any benefit given that SGD seems to find good solutions in the overparametrized case? 

7. **Authors have answered this in the revision** Experiments: exp. 5.2. same as before: what happens with the misclassification error? what stops you from doing the computation on all the data? memory?
Does the red cross mean that the soft thresholded SVD is solved in less than one second? 

References:
1. Unifying Nuclear Norm and Bilinear Factorization Approaches for Low-Rank Matrix Decomposition
Ricardo Cabral, Fernando De La Torre, Joao P. Costeira, Alexandre Bernardino; Proceedings of the IEEE International Conference on Computer Vision (ICCV), 2013, pp. 2488-2495
2. Geometry of Factored Nuclear Norm Regularization. Qiuwei Li, Zhihui Zhu, Gongguo Tang
3. Guaranteed Minimum-Rank Solutions of Linear Matrix Equations via Nuclear Norm Minimization. Benjamin Recht, Maryam Fazel, and Pablo A. Parrilo

+ many references therein

overall I would be happy to recommend acceptance if the previous issues were adressed in a succint way.

**Update** After author feedback many of my concerns have been addressed, in particular the optimization problem
templates are much easier to understand now as well as the derivation of the algorithm. Also some important differences
between linear classifiers vs ReLU networks on 'Spike-free' data have been clarified. These were my main concerns and thus I am inclined to raise my score.

---

> ### Author Response · Authors · 2020-11-16
> **Thank you for the review. (see cont.)**
>
> We thank the reviewer for their comments and suggestions. We have cleared up many of the issues that the reviewer would like to have addressed. Please see our point-to-point response below.
>
> 1. "Clarity: I could not really understand the optimization problem (9) and (11) because the variable i appears as an optimization variable  mini∈[P] but it does not appear in the optimization objective. "
>
> Yes, this was a typo. We have corrected the cases where this error occurred in the revised version. Thanks for pointing this out.
>
> 2. "Clarity: It is not so clear how Algorithm 1 follows the Frank-Wolfe template."
>
> We should make more clear that $t$ is fixed throughout the duration of the Frank-Wolfe iterations, which only solve the inner minimization problem of (11). As an outer loop, the optimal value of $t$ can be found via bisection. Indeed the step (a) computes the LMO step. We have derived this LMO step, and the entire Frank-Wolfe algorithm in much more detail from first principles in Appendix A.4.1.
>
> 3. "Significance: The Spike-free assumption seems to simplify things but it might be unrealistic... Does whitened X imply spike-free X?"
>
> Thank you for mentioning this point. As we have mentioned in our response to Reviewer 4 as well, the spike-free assumption is not exactly identical to a linear model--in particular, there is the additional constraint that $$\boldsymbol{V} \in \mathbf{conv}\Big(\boldsymbol{u}\boldsymbol{g}^\top: \boldsymbol{X} \boldsymbol{u} \geq 0, \|\boldsymbol{g}\|_2 \leq 1\Big) $$
> This constraint is quite significant, as it is the difference between standard PCA and semi-Nonnegative PCA (for example, consider the steps in Algorithm 1 in the case where the constraint is not present as would be the case in a linear-activation network, and compare it to when the constraint is present). This means that the ReLU allows for a much richer and complex representation than the linear classifier, even in the case of the spike-free assumption. In appendix A.1.2 we have provided an experiment which demonstrates that even in the spike-free case, the ReLU-activation network and linear-activation network perform quite differently.
>
> The spike-free assumption does provide some simplifications in the analysis, but this is not the only reason which we consider this. In particular, the spike-free assumption makes an important distinction between scalar-output and vector-output NNs clear, namely spike-free data matrices make training scalar-output NNs tractable in polynomial time, but with the same data setting, training vector-output NNs is tractable in polynomial time only if P=NP (see Table 1), which is unlikely. This suggests that there is something fundamentally harder about training vector-output NNs as opposed to scalar-output NNs.
>
> Whitening is also something that has been used to improve neural network training, in particular batch whitening. See for example (Huang et al, 2018), which batch-wise performs ZCA whitening as a replacement for Batch Normalization and finds large improvements in performance in deep networks. Thus, it is not a given that the spike-free assumption or whitened data assumption is unrealistic. We have included a minor justification of the spike-free setting towards the end of section 2 in our revised paper. We made sure to cite additional relevant literature regarding the nuclear-norm regularization as well, such as after Theorem 2.
>
> And yes, whitened X implies spike-free X whenever $n \leq d$, which was discussed towards the end of section 2. We have reiterated this point in the revised version prior to the statement of Theorem 2.
>
> 4. "Significance: Theorem 2 is a simple consequence of this "spike-free" simplified setting and corresponds to the solution of the proximal operator of the nuclear norm."
>
> In Theorem 2, the additional assumption on the left-singular vectors is what ensures the solution of the proximal operator of the nuclear norm. To this end, we have made this point more clear in our revised version--in this case, the ReLU network is indeed identical to a linear-activation network. This is still a novel result--it outlines the setting in which the ReLU network is identical to a linear-activation network, and provides a simple closed-form solution in this setting.
>
> **References**
> Lei Huang, Dawei Yang, Bo Lang, and Jia Deng. Decorrelated batch normalization. *In
> Proceedings of the IEEE Conference on Computer Vision and Pattern Recognition*, pp. 791–800, 2018.

---

> > ### Author Response · Authors · 2020-11-16
> > **Thank you for the review (cont.)**
> >
> > 5." Significance: the previous two points suggest that the important results here are those corresponding to general X matrix...It looks like the proposed algorithms are currently only of theoretical interest."
> >
> > Yes, this is an important point. The complexity for general X matrices is indeed exponential in the rank of the data, but as we have discussed in depth in the paper (such as in Remark 1.1 or in the caption of Table 1), for CNNs with a fixed filter-size, the effective rank of the data matrix is fixed, so the convex program is polynomial in the number of data points and the size of the images. Note that modern CNNs have fixed filter sizes, e.g., with m filters of size 3x3 the complexity is polynomial in all parameters (m,n,d). Thus, the proposed algorithms are quite tractable for these settings.
> >
> > This is also a fundamental limit, as we show that solving the neural network training problem is equivalent to solving an NP-hard problem, which must be exponential time in the dimension unless P=NP. This is indeed an important theoretical result, but one which can be applied practically in certain scenarios.
> >
> > 6. "Experiments: exp 5.1… compared to SGD do you think there is any benefit given that SGD seems to find good solutions in the overparametrized case?"
> >
> > The labels for this experiment are continuous-valued, so this is not a classification problem for which we can measure the 0-1 error. However, for experiment 5.2, we can and have done so (please see the response to the next point). Compared to SGD, the main benefit of the Frank-Wolfe algorithm is that it is guaranteed to find a global solution to the NN training problem for $m \geq m^*$, whereas no such guarantee occurs for SGD. The under-parameterized case is just one such example in which SGD fails to find the global minimum--there may be other pathological cases.
> >
> > Currently, the convex problem in experiment 5.1, we run for 30000 Frank-Wolfe iterations, and it takes 30 minutes to solve--however we should note that this is with a naive implementation, which has not been optimized for scalability and does not use GPU.
> >
> > 7. "Experiments: exp. 5.2. same as before: what happens with the misclassification error?... Does the red cross mean that the soft thresholded SVD is solved in less than one second?"
> >
> > The test misclassification error is shown in Figure 6 in Appendix A.1.4. The misclassification error is identical between the solutions found by the dual program and SGD. This experiment is to evaluate the setting of Theorem 2, which means that we require a whitened data matrix for which $n \leq d$. Thus, we require there to be $\le d= 3072$ data points to evaluate this setting. Indeed, the red cross indicates that the soft-thresholded SVD is solved in less than one second--approximately 0.018 seconds for CIFAR-10, and 0.36 seconds for CIFAR-100-- we have also added these details to the appendix.

---

> > > ### Comment · AnonReviewer1 · 2020-11-23
> > > **thanks for the feedback**
> > >
> > > I have updated my review. I think the new version clarifies many things and some of my concerns have been addressed. I think this paper allows further interesting discussion and thus would be valuable to the community. I have raised my score.

---

### Official Review · AnonReviewer3 · 2020-10-29
**Incremental generalization of recent results in the convexification of neural networks to vector outputs**

**Rating:** 7
**Confidence:** 3

**Review:**

## Summary
The paper proposes a convex formulation for shallow neural networks with one hidden layer and vectorial outputs. This is an extension on a line of previous works (Ergen & Pilanci, 2020a) and (Ergen & Pilanci, 2020b) where similar results have been established for the case of scalar outputs. A Frank-Wolfe algorithm for finding the global optimum of the resulting convex program is proposed and evaluated on smaller datasets.

## Explanation of Rating
Overall, the paper tackles an important problem, is technically sound and well-written. I recommend acceptance of this paper, but have some smaller doubts which are mainly due to the incremental nature of the results (see detailed comment #1) and the strong assumptions for the proposed relaxation to be exact (see comment #2).

## Detailed Comments
1. My main concern with the paper is, that it is a direct extension of the results from (Ergen & Pilanci, 2020a) and (Ergen & Pilanci, 2020b) to the vectorial setting and therefore lacks novelty. Therefore, I feel the results presented in this paper might be better suited for a journal version of these previous works.

2. The tightness guarantee for the relaxation requires the number of hidden neurons to be larger than the number of data points. In practice, this assumption is often not satisfied. The paper would be stronger if there was some theory (e.g. a duality gap analysis) that quantifies how the quality of the relaxation degrades as the number of data points is increased.

---

> ### Author Response · Authors · 2020-11-16
> **Thank you for the review.**
>
> We thank the reviewer for their helpful review. We would like to respond to their comments and suggestions below.
>
> 1. "It is a direct extension of the results from (Ergen & Pilanci, 2020a) and (Ergen & Pilanci, 2020b) to the vectorial setting and therefore lacks novelty."
>
> While the setting of the work is similar to those of (Ergen & Pilanci, 2020a) and (Ergen & Pilanci, 2020b), the analysis we perform is quite different from those in that work. In particular, we are faced with additional theoretical challenges in finding the dual problem, in which we require taking a convex hull of a set which is difficult to characterize, and we draw new connections to copositive programs, semi-NMF, and cone-constrained PCA, which were not seen in those works. We also demonstrate fundamental differences in computational complexity from the scalar-output case, wherein even with a spike-free data matrix the problem is NP hard in the vector-output case, while in the scalar-output case it is polynomial time.
>
> 2. "The tightness guarantee for the relaxation requires the number of hidden neurons to be larger than the number of data points. In practice, this assumption is often not satisfied. The paper would be stronger if there was some theory (e.g. a duality gap analysis) that quantifies how the quality of the relaxation degrades as the number of data points is increased."
>
> This is a subtle point, but we should be clear that the assumption does not require the number of hidden neurons $m$ to be greater than the number of data points $n$. As stated in Theorem 1, our only condition is $m \geq m^*$, for an unknown $m^*$ which is upper bounded by $nc +1$, where $c$ is the output dimension. In practice $m^*$ is much smaller than $nc$--for example in our first experiment 5.1, we can deduce that $10 \leq m^* \leq 50$, since the duality gap shrinks to zero when the number of neurons increases from 10 to 50. In general, $m^*$ can be determined by solving the convex dual and observing how many dual neurons are non-zero at optimum.
>
> For the $m < m^*$ case, this is a good point. Our analysis does offer an intuition into measuring the duality-gap empirically. One way to measure the duality gap would be to simply solve the dual problem, and then greedily remove the least impactful dual neurons until only $m$ of them remain. The duality gap would then simply be the difference between the new loss with $m$ neurons and the original loss with $m^*$. However, a general form of the duality gap is likely intractable to find, because it is so dependent on the data \(X\) and \(Y\).

---

### Official Review · AnonReviewer4 · 2020-10-30
**Good extension, but studying the whiten case is not useful (concerns answered in the author feedback).**

**Rating:** 7
**Confidence:** 2

**Review:**

The draft is a vector extension of [1] on studying how to approximately solve the global optima of a two-layered Relu network. The key of the analysis is to enumerate all possible sign patterns of the ReLU unit generating from specific data. Once we have the enumeration, we can also enumerate the linear area separated by ReLU, and the whole optimization problem will become a non-convex quadratic optimization problem. The non-convex quadratic can be approximately solved with its convex dual (or exactly under some conditions), or we can relax it to a copositive program (which might still be NP-hard to solve). With some assumption on the data, the sign pattern of the ReLU is a singleton, then we will have efficient algorithms to exactly recover the global optima of the two-layered network.

I like the idea of linking 2-layer NN with copositive programs, the writing is good, and the proofs seem to be correct.
However, I feel that the study on the whiten case with a singleton ReLU sign pattern is really not useful. When all the sign patterns of the ReLU are pinned, the representation power of a two-layered network reduces to a linear model (in a polyhedron). And if the representation power is equivalent to a linear model, why not reparameterize it linearly? The specific case lost all the existential meaning for a ReLU unit. (prove me wrong by showing it is better than linear in experiment.)

Overall, this is a good extension of [1], but the improvement in complexity only works in trivial case.

Minor issues:
1. (Algorithm 1, (b)) The uj gj is undefined. The sj^(k) is not used anywhere.
2. (equation at the bottom of p.18) Misisng sqrt{}
3. Compleity below Section 4.2: enumeration takes O(n^r), but solving a copositive program is still NP.
4. (Algorithm 1) It takes me a while to realize how the Frank-Wolfe algorithm works for the $\sum_i|Vi|\leq t$ constraint. Maybe it can be more clear.
5. (Writing) There are many reference to appendix without specific section number... Please add them.

[1] Neural Networks are Convex Regularizers: Exact Polynomial-time Convex Optimization Formulations for Two-layer Networks. Mert Pilanci and Tolga Ergen

---

> ### Author Response · Authors · 2020-11-16
> **Thank you for the review.**
>
> We thank the reviewer for their comments and suggestions. Please see our point-to-point response below.
>
> 0.  "I feel that the study on the whiten case with a singleton ReLU sign pattern is really not useful. When all the sign patterns of the ReLU are pinned, the representation power of a two-layered network reduces to a linear model (in a polyhedron)... prove me wrong by showing it is better than linear in experiment."
>
> We would like to respectfully disagree. As this reviewer has mentioned, the spike-free assumption is not identical to a linear model--in particular, there is the additional constraint that
> $$\boldsymbol{V} \in \mathbf{conv}\Big(\boldsymbol{u}\boldsymbol{g}^\top: \boldsymbol{X} \boldsymbol{u} \geq 0, \|\boldsymbol{g}\|_2 \leq 1\Big) $$
> i.e. the polyhedron mentioned. This polyhedron is quite significant for the representation power of the network. In particular, (12) without the constraint is indeed identical to a linear model. However, the constraint makes the problem of cone-constrained PCA, which allows for modeling of much more complex functions. This means that the ReLU allows for a much richer and complex representation than the linear classifier, even in the case of the spike-free assumption. We have made this distinction more clear in Section 3.3.
>
> As an example, we performed a small-scale experiment to differentiate between ReLU and linear-activation networks when the spike-free assumption holds. The details of this experiment can be found in appendix A.1.2. Ultimately, we see that for the dataset considered, the ReLU network performs better than the linear network across all values of  $\beta$, as we would expect. This suggests that even in the spike-free case, the ReLU still has an important role.
>
> 1. "(Algorithm 1, (b)) The uj gj is undefined. The sj^(k) is not used anywhere."
>
> We have made the definition of (u_i, g_i) clear as the arg max from the Frank-Wolfe subproblem for each i. We also now no longer use sj^(k)  in Algorithm 1.
>
> 2. "(equation at the bottom of p.18) Missing sqrt{}"
>
> For simplicity, we had removed the sqrt{}, because squaring the objective does not change the optimal value or the ordering of the $s_i^{(k)}$, since the term in the objective is non-negative. We believed that the problem without the square-root would be easier to link directly to the cone-constrained PCA problem. We have made this more clear in the revised version (see appendix A.4.2).
>
> 3. "Complexity below Section 4.2: enumeration takes O(n^r), but solving a copositive program is still NP."
>
> Section 4.2 is no longer precisely a copositive program, since we relax the copositivity constraint. The problem presented in (16) is indeed NP, but this is because of the enumeration over sign patterns. Once we reduce to a single sign pattern such as in the spike-free case of Section 4.3, the problem is no longer NP. In short, either spike free X or copositive relaxation alone does not improve upon NP-hardness, but using both of them together does.
>
> 4. "(Algorithm 1) It takes me a while to realize how the Frank-Wolfe algorithm works for the ∑i|Vi|≤tconstraint. Maybe it can be more clear."
>
> We have derived how the Frank-Wolfe steps in Algorithm 1 were obtained in a much more explicit manner in appendix A.4.1. We hope that how it is explained now is clear.
>
> 5. "(Writing) There are many reference to appendix without specific section number... Please add them."
>
> Yes, thank you for this point. We have updated references to the appendix by their section number.

---

> > ### Comment · AnonReviewer4 · 2020-11-24
> > **Still not convinced with the usefulness**
> >
> > I'm satisfied with the corrections in the minor comments, but I'm still not convinced by the additional experiments in Appendix A.1.2,
> > because it has the following flaws:
> >
> > 1. By ZCA, your data is already normalized. In your Jupyter notebook, you take a 3k subset of the CIFAR-10, training it on 4k neurons, and it appears that you have an average testing error 10^5 to 10^9? The error itself is even larger than the norm of the dataset. Clearly, there's something wrong here.
> > 2. I ran your code and checked the accuracies. In the case that the testing error of ReLU is significantly lower than a linear network (still unreasonable number), the linear network has higher accuracies. Take beta=0.001 as an example, the ReLU network has a 2.8e3 testing loss (better than your report) and a 35% accuracy, and the linear network has a 1.7e7 loss and 41% accuracy. So you cannot claim that after whitening, the ReLU consistently generalizes better than the linear network.
> > 3. In your code, you set the batch size to be the number of data, so it is running a GD instead of an SGD. It is possible that both models are not well-trained. BTW, for the linear network, you can actually easily obtain the optimal weight by solving the linear equation (on 3k dimension).
> > 4. I checked the sign patterns in the ReLU units. They are not a singleton in the experiment, as opposed to the assumption of the paper. May you explain this?
> >
> > Overall, I'm still not convinced that a ReLU network with a singleton sign-pattern can outperform a linear network. It is still interesting work, but I will not change my score.

---

> > > ### Author Response · Authors · 2020-11-25
> > > **Further Clarifications on Spike-Free Scenario**
> > >
> > > There are a few significant misunderstandings with regards to the additional experiment we have added to the appendix to distinguish between spike-free ReLU-activation and linear models. Please see our point-to-point response below for a detailed explanation.
> > >
> > > 1. “By ZCA, your data is already normalized… The error itself is even larger than the norm of the dataset. Clearly, there's something wrong here.”
> > >
> > > To be clear, we ZCA whiten the ***training*** data, and use the same transformation to apply to the test data as well. There is no reason why the test error should necessarily be less than the norm of the test labels, especially when we are performing inference from so few training examples.
> > >
> > > 2. “I ran your code and checked the accuracies. In the case that the testing error of ReLU is significantly lower than a linear network (still unreasonable number), the linear network has higher accuracies... So you cannot claim that after whitening, the ReLU consistently generalizes better than the linear network.”
> > >
> > > What labels are you using to check for accuracy? In particular, in the experiment in Appendix A.1.2, we use continuous-valued labels generated by a neural network, so accuracy has no meaning in this circumstance. If instead you mean that you trained this model for the standard classification task using squared loss and one-hot encoded labels, this result you describe is indeed possible.
> > >
> > > The point of this experiment is not to show that ReLU-activation networks always outperform linear-activation networks for spike-free data. Just as for non-spike-free data, there may be applications where linear-activation networks perform better than ReLU-activation networks.
> > >
> > > The point of our experiment is to show that even in the spike-free setting, linear-activation and ReLU-activation networks are quite different, and ReLU-activation networks *can* generalize better, with the experiment in Appendix A.1.2 being one such example. This is in response to the earlier point that “The specific case lost all the existential meaning for a ReLU unit.” The point is that in the spike-free case, a linear-activation model is not identical to a ReLU-activation model, so analyzing this setting still provides insight into the ReLU regime.
> > >
> > > 3. “In your code, you set the batch size to be the number of data, so it is running a GD instead of an SGD. It is possible that both models are not well-trained. BTW, for the linear network, you can actually easily obtain the optimal weight by solving the linear equation (on 3k dimension).”
> > >
> > > Yes, this is a good point. We can easily use the soft-thresholded SVD solution from Theorem 2 for the linear-activation network. In practice if you try this method you will find that it nearly coincides with the solution from GD and does not generalize as well as the ReLU network. For the ReLU network there is no guarantee that GD or SGD will find the global optimum in this case, but this was just intended to be a simple example to show that ReLU can perform better than linear activation even in the spike-free case. Either way it should be clear that the two models are quite different, because of the polyhedron constraint as described in the paper and reiterated in our previous response.
> > >
> > > 4. "I checked the sign patterns in the ReLU units. They are not a singleton in the experiment, as opposed to the assumption of the paper. May you explain this?"
> > >
> > > Indeed, the sign patterns for spike-free data will not be a singleton, but the optimization problem is *as if there were only a single sign pattern*. To clarify this point, consider a simple example.
> > >
> > > Let $X = [1/\sqrt{2}, 1/\sqrt{2}; 1/\sqrt{2}, -1/\sqrt{2}]$. Clearly this matrix is spike-free. The sign patterns are set by $\mathcal{D} = \Big( \mathrm{diag}(1_{Xu\ge0}) : \|u\|_2 \le 1\Big)$. Consider the vectors $u_1 = [1; 0]$, $u_2 = [1/\sqrt{5}; 2/\sqrt{5}]$, $u_3 = [-2/\sqrt{5}; 1/\sqrt{5}]$,  $u_4 = [-1/\sqrt{5}; -2/\sqrt{5}]$, which correspond to sign patterns $D_1 = [1, 0; 0, 1]$, $D_2 =[1, 0; 0, 0]$,  $D_3 = [ 0, 0 ;0 , 0]$, $D_4 =[0, 0; 0, 1]$.
> > >
> > > Clearly, the set of sign patterns of X is thus not a singleton. However, the following equality still holds: $\Big( (Xu)_+ : \|u\|_2 \le 1 \Big) = \Big(Xu : \|u\|_2 \le 1\Big) \cap \mathbb{R}^{2+}$. This is proven from Lemma 2.3, 2.4 of (Pilanci & Ergen, 2020).
> > >
> > > This allows us to simplify the set $\Big((Xu)_+ : \|u\|_2 \le 1\Big) $ with  $\Big(Xu : \|u\|_2 \le 1, Xu \ge 0 \Big) $, which is identical to the singleton sign pattern $D_1 = I$, which allows for the simplification done in the spike-free sections of the paper. This is explained in page 4 of the paper and in further detail in (Pilanci & Ergen, 2020). Please refer to these for clarity on this point.
> > >
> > > **References**
> > >
> > > Tolga Ergen and Mert Pilanci. Convex geometry and duality of over-parameterized neural networks. *arXiv preprint arXiv:2002.11219*, 2020.

---

> > > > ### Comment · AnonReviewer4 · 2020-11-25
> > > > **Concerns answered in the author feedback.**
> > > >
> > > > The feedback explained my main concern. Now I understand it is not that D is a singleton for the original ReLU network, but that there is an equivalent weight $u$ in a linear network realizing the same sign pattern (requiring n<=d) while satisfying $Xu\geq 0$. I think this is not very clear in the current writing. Please clarify this in the revision.
> > > >
> > > > BTW, for point 2, I got the accuracies using your code (return_acc), which applies argmax on the continuous outputs. In those cases, the linear network does have better accuracies than the ReLU network for the whitened data.

---

### Author Response · Authors · 2020-11-16
**Summary of Our Response to Reviewers**

We thank all the reviewers for their helpful comments and suggestions. We believe that we have significantly improved the paper to address the concerns of the reviewers, for which we uploaded the revised version. In summary, our main changes were as follows:

1. Included further clarification on Algorithm 1 by making the notation more clear and by deriving the Frank-Wolfe update steps in Appendix A.4.1.
2. Made more clear the distinction between ReLU- and linear-activation networks in the case of spike-free data matrices, both in the discussion in Section 3.3, and with an additional experiment in Appendix A.1.2.
3. Connected to other significant related work in both neural network theory and nuclear norm minimization, including but not limited to (Ge et al 2018), (Brutkus & Globerson, 2017), (Recht et al., 2010), (Cabral et al., 2013), (Li et al., 2017), and (Huang et al, 2018).
4. Fixed equation typos (such as in (9) and (11)) and improved references to the appendix.
5. Added points to the conclusion about how our work may be extended to deeper networks.

We have also responded to each reviewer individually to address their points.

---

### Decision · Program_Chairs · 2021-01-07
**Final Decision**

**Decision:**

Accept (Poster)

**Comment:**

This paper extends an earlier work with scalar output to vector output. It establish a relationship of two-layer ReLu network and convex program. The result can be used to design training algorithms for ReLu networks with provably computational complexity. Overall, this is an interesting idea, leading to better theoretical insights to computational issues of two-layer ReLu networks.